# Robust AI Evaluation through Maximal Lotteries

**Hadi Khalaf** [1]   **Serena L. Wang** [† 1]   **Daniel Halpern** [† 1]   **Itai Shapira** [† 1]   **Flavio P. Calmon** [‡ 1]   **Ariel D. Procaccia** [‡ 1]

## Abstract

The standard way to evaluate language models on subjective tasks is through pairwise comparisons: an annotator chooses the "better" of two responses to a prompt. Leaderboards aggregate these comparisons into a single Bradley-Terry (BT) ranking, forcing heterogeneous preferences into a total order and violating basic social-choice desiderata. In contrast, social choice theory provides an alternative approach called *maximal lotteries*, which aggregates pairwise preferences without imposing any assumptions on their structure. However, we show that maximal lotteries are highly sensitive to preference heterogeneity and can favor models that severely underperform on specific tasks or user subpopulations. We introduce robust lotteries that optimize worst-case performance under plausible shifts in the preference data. On large-scale preference datasets, robust lotteries provide more reliable win rate guarantees across the annotator distribution and recover a stable set of top-performing models. By moving from rankings to pluralistic sets of winners, robust lotteries offer a principled step toward an ecosystem of complementary AI systems that serve the full spectrum of human preferences.

## 1. Introduction

Evaluating large language models (LLMs) on open-ended, subjective tasks is intrinsically challenging. Whereas tasks such as mathematics and coding often have objective success criteria, a significant portion of LLM use cases, including seeking information, practical guidance, and writing (Chatterji et al., 2025), are difficult to assess because they can lack a single ground truth and reliable metrics (Zheng et al., 2023; Zhou et al., 2024; Liu et al., 2024). In such settings, a common approach to evaluation is to rely on pairwise comparison data (Stiennon et al., 2020; Chiang

et al., 2024), where an annotator (either human or AI) selects the preferred of two model outputs for a given prompt. Aggregating these judgments across prompts and annotators yields a single global summary, often a score or ranking.

AI leaderboards such as LMArena (Chiang et al., 2024) and MedArena (Wu et al., 2024) construct rankings from pairwise comparisons by fitting a one-dimensional random utility model, most commonly the Bradley–Terry (BT) model (Bradley & Terry, 1952). Such models assign each system a single latent "strength" parameter, which is mathematically convenient but cannot capture the multi-faceted, prompt-dependent nature of LLM performance. Instead, they force diverse, context-dependent judgments into a single ordering – despite significant evidence that preferences are often non-transitive in LLM settings (Zhang et al., 2025a; Swamy et al., 2024; Zhang et al., 2025b). This misspecification can make BT yield counterintuitive results: even when a model is preferred to every other model in pairwise comparisons, the fitted BT ranking need not place it first (Lee & Chen, 2025). The BT framework also violates clone invariance, where duplicating a model can change the rankings in arbitrary ways (Tideman, 1987; Lanctot et al., 2025).

These issues motivate using aggregation rules designed for general pairwise preference data, without assuming transitivity or a one-dimensional score. Social choice theory (Brandt et al., 2016) offers principled alternatives, including rules that output either a ranking or a set of winners (Brandt et al., 2016; Ge et al., 2024; Conitzer et al., 2024; Dai & Fleisig, 2024; Halpern et al., 2025).

We focus on *maximal lotteries* (Fishburn, 1984; Brandl et al., 2016). Consider a majority margin matrix whose entries record the extent to which one model is preferred to another; see Definition 1. Maximal lotteries treat the empirical majority margins as a symmetric zero-sum game and output a maximin mixed strategy. The result is a distribution $p$ over models such that a model sampled from $p$ does not lose in expectation against any fixed model $j$. This guarantee remains meaningful under non-transitive preferences because it does not require a globally consistent ranking. Maximal lotteries address many limitations of BT by making no assumptions about the underlying structure of preferences and by being invariant under the addition of clones. For this reason, Lanctot et al. (2025) identified maximal lotteries as

---

[†‡]Listed in randomized order. [1]Harvard University. Correspondence to: Hadi Khalaf <hadikhalaf@g.harvard.edu>.

*Proceedings of the 43rd International Conference on Machine Learning*, Seoul, South Korea. PMLR 306, 2026. Copyright 2026 by the author(s).

a promising alternative for comparing generative models. Unlike traditional leaderboards, maximal lotteries do not output a ranking. Rather, they identify a competitive set of models, called the *bipartisan set* (Laffond et al., 1993). This set is a natural notion of a "top tier" under the observed preference data. The lottery formulation is also operationally flexible as we discuss in Section 4.2. It allows incorporating practical constraints on the fly, such as cost budget or benchmark performance, to identify the set of most performant models.

Despite fixing key misspecification issues in BT, maximal lotteries introduce a different failure mode: they can be brittle to shifts in the prompt mix or annotator population. As a result, maximal lotteries can yield bipartisan sets that are not *pluralistic*: a set of models can seem optimal on preference data aggregated across an entire population, yet have large worst-case performance gaps when evaluated on preferences from specific subpopulations.

Consider, for example, Figure 1. To analyze how diverse preferences impact model evaluation, we partition data from LMArena based on the language of the prompts provided by annotators. The standard maximal lottery (orange) optimizes for the aggregate preference distribution and results in a bipartisan set with a single model, Gemini 2.5 Pro. While this model is the strongest *on average*, it leaves a significant performance gap with respect to some languages. In contrast, the *robust lottery* (blue) identifies a distribution across four distinct models – Gemini 2.5 Pro, DeepSeek R1, Llama 4 Maverick, and Opus 4 – that balance the inherent tradeoffs across subpopulations. By including more models in the bipartisan set, robust lotteries remain reliable across the entire annotator distribution.

**Contributions.** Our main contributions include:

1. **Robust lotteries.** We introduce robust lotteries that maximize worst-case win probability over an ambiguity set of majority-margin matrices, without assuming transitivity or latent utilities.

2. **Axiomatic and structural properties.** We show that robust lotteries satisfy desirable social choice guarantees, including weak-clone invariance and a robust Condorcet-style consistency property.

3. **Tractable computation.** For uncertainty over subpopulation weights, we derive an efficient linear program and provide finite-sample regret bounds.

4. **Empirical robustness.** On four language model benchmarks, robust lotteries improve worst-case win rate guarantees and recover a set of frontier models.

## 2. Related Work

**Aggregating pairwise preferences.** Pairwise preference modeling has a long history in psychology, statistics, and social choice, with Bradley-Terry and Thurstone models as canonical examples (Bradley & Terry, 1952; Thurstone, 2017; Luce, 1959; Plackett, 1975). The rise of LLMs has made pairwise preferences central to AI–human alignment (Stiennon et al., 2020; Christiano et al., 2023; Bai et al., 2022; Ouyang et al., 2022) and to large-scale evaluation (Chiang et al., 2024), motivating methods that improve their efficiency and reliability (Ameli et al., 2025; Verma et al., 2025; Liu et al., 2025b;a; Xu et al., 2025; Dubois et al., 2025). At the same time, user- and task-dependent preferences often violate a single total order, which calls for more general aggregation objectives (Swamy et al., 2024; Zhang et al., 2025b). In particular, AI leaderboards collapse model evaluation into a score that implicitly fixes how users and tasks are weighted, so rank orderings can shift arbitrarily under reasonable changes to that weighting or to the evaluation protocol itself (Dehghani et al., 2021; Boubdir et al., 2024; Siska et al., 2024; Zhang & Hardt, 2024). We therefore propose an evaluation method that remains reliable under a set of plausible margin matrices, without making any assumptions on the collected preferences themselves.

**Robust game theory and robust optimization.** Our formulation of robust lotteries includes a minimax objective which is closely related to the *robust-optimization equilibrium* introduced by Aghassi & Bertsimas (2006) in which both players maximize their payoff over an uncertainty set of payoff matrices. However, in our setup, only one "player" operates under uncertainty, making our exact formulation closer to a robust Stackelberg game (Kroer et al., 2018). Similarly, Jeyakumar et al. (2011) study the existence of minimax equilibria under norm-bounded payoff perturbations, whereas our uncertainty is structured through reweighting subpopulation-specific margin matrices. To our knowledge, such formulations have not previously been applied in the context of maximal lotteries. Moreover, the setup of our ambiguity set draws from the literature on distributionally robust optimization (Duchi & Namkoong, 2021), in which the goal is to train machine learning models that perform well under covariate shift. Thus, we combine a game theoretic foundation with ideas from distributionally robust optimization to build maximal lotteries that are robust to shifts in annotator and prompt distributions.

**Maximal lotteries.** Maximal lotteries were first proposed by Kreweras (1965) and then studied by Fishburn (1984). Unlike any deterministic social choice rule, maximal lotteries can simultaneously satisfy demanding axioms, such as consistency across varying electorates and invariance to clones (Brandl et al., 2016). As such, these methods have been repeatedly advocated as decision procedures over the decades (Brandl et al., 2022; Zeckhauser, 1969; Stone,

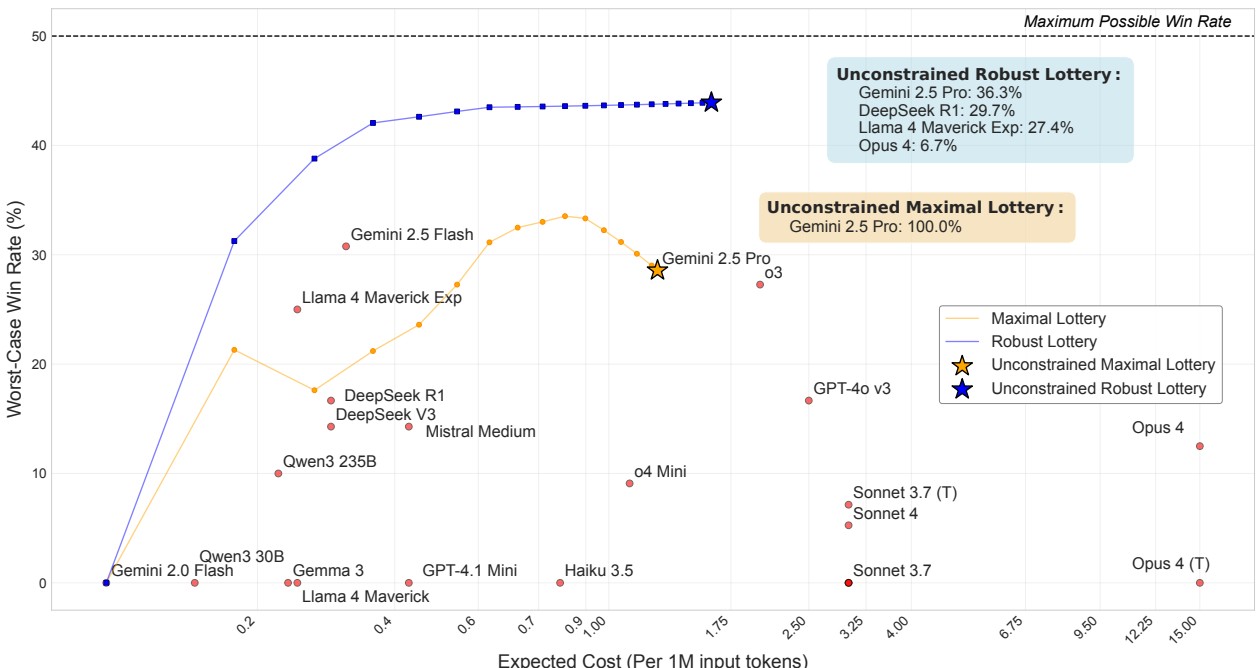

*Figure 1.* **Robust lotteries improve worst-case win rate across user subpopulations.** We consider a fixed set of models (red points) and partition the LMArena votes according to the language of the prompt. We then evaluate any lottery (a probability distribution over models) by its *worst-case performance* on LMArena (*y*-axis): the minimum win rate the lottery achieves across the four user subpopulations. We also present the performance on the overall preference matrix in Figure B.3. Because the guarantee is $\min_q \Pr(p \succ q)$ in a symmetric zero-sum game, it is at most 50%. The *x*-axis reports the policy's expected inference cost per 1M input tokens. The **orange** curve traces the cost–performance frontier obtained by solving for the *maximal lottery* under an expected-cost budget. The **blue** curve traces the corresponding frontier for *robust lotteries*, which instead optimize the worst-case guarantee using a robust linear program. Across budgets, robust lotteries achieve substantially higher worst-case performance than the standard maximal lottery.

2011). Beyond social choice, similar constructions have been rediscovered many times under different names, e.g., the "game theory procedure" in voting (Rivest & Shen, 2010) and the von Neumann winner (Dudík et al., 2015; Swamy et al., 2024).

## 3. Background on Maximal Lotteries

In this section, we define maximal lotteries and state some of their key properties. We refer the reader to Fishburn (1984) and Brandl et al. (2016) for further details.

We encode pairwise comparison data as a majority-margin matrix and treat it as the payoff matrix of a symmetric zero-sum game. A *lottery* over models is then a mixed strategy for this game, and a *maximal lottery* is the strategy that maximizes its worst-case margin against any opponent mixture.

**Setup.** Let $A = \{1, \ldots, m\}$ be a set of $m$ alternatives (e.g., models). Let $\Delta(A) = \{p \in \mathbb{R}_{\geq 0}^m : \sum_{i=1}^m p_i = 1\}$ denote the probability simplex over $A$. For $p \in \Delta(A)$, define the support $\mathrm{supp}(p) = \{i \in A : p_i > 0\}$. Let

$$\mathbb{M}_m = \{M \in [-1, 1]^{m \times m} : M = -M^\top\}$$

be the space of $m \times m$ skew-symmetric matrices with entries bounded in $[-1, 1]$. We observe pairwise comparisons. For each ordered pair $(i, j)$, let $w_{ij}$ be the number of comparisons in which model $i$ is preferred to model $j$. Let $n_{ij} = w_{ij} + w_{ji}$ denote the total number of comparisons between $i$ and $j$.

**Definition 1** (Majority Margin Matrix). The majority margin matrix $M \in \mathbb{M}_m$ is defined entrywise by

$$M_{ij} := \begin{cases} \frac{w_{ij} - w_{ji}}{n_{ij}} & \text{if } n_{ij} > 0, \\ 0 & \text{if } n_{ij} = 0. \end{cases}$$

The entry $M_{ij}$ is the empirical win margin of alternative $i$ against $j$.

**Definition 2** (Maximal Lotteries (Fishburn, 1984)). A *maximal lottery* is a distribution $p^\star \in \Delta(A)$ such that:

$$p^\star \in \arg\max_{p \in \Delta(A)} \min_{q \in \Delta(A)} p^\top M q.$$

Let $\mathrm{ML}(M)$ denote the set of maximal lotteries for $M$. Because $M$ is skew-symmetric, the maximin value is 0. In particular, any maximal lottery satisfies

$$p^{\star\top} M q \geq 0 \quad \text{for all } q \in \Delta(A), \tag{1}$$

and there exists at least one best response $q^\star$ such that $p^{\star\top} M q^\star = 0$.

**Definition 3** (Bipartisan Set (Laffond et al., 1993))**.** The *bipartisan set* is the set of alternatives that receive positive probability in at least one maximal lottery:

$$\mathrm{BP}(M) := \bigcup_{p \in \mathrm{ML}(M)} \mathrm{supp}(p).$$

Maximal lotteries admit a direct interpretation in terms of win rates. Define the induced pairwise win rate by

$$\Pr(i \succ j) := \tfrac{1}{2} + \tfrac{1}{2} M_{ij}.$$

For two lotteries $p, q \in \Delta(A)$, sample $I \sim p$ and $J \sim q$. Then

$$\begin{aligned} \Pr(p \succ q) &:= \Pr(I \succ J) \\ &= \sum_{i,j} p_i q_j \Pr(i \succ j) = \tfrac{1}{2} + \tfrac{1}{2} p^\top M q. \end{aligned}$$

If $p^\star$ is a maximal lottery, then Equation (1) implies $\Pr(p^\star \succ q) \geq \tfrac{1}{2}$ for all $q \in \Delta(A)$. Equivalently, no fixed model and no mixture of models can beat $p^\star$ in expectation.

**Example.** Let $A = \{1, 2, 3\}$ and consider two user subpopulations defined by their primary language. Each group induces a margin matrix as in Definition 1. In EN, let

$$M^{(\mathrm{EN})} = \begin{pmatrix} 0 & 0.6 & 0.6 \\ -0.6 & 0 & 0.6 \\ -0.6 & -0.6 & 0 \end{pmatrix},$$

so Model 1 beats both others and the maximal lottery is the point mass on Model 1. In ES, let

$$M^{(\mathrm{ES})} = \begin{pmatrix} 0 & 0.6 & -0.6 \\ -0.6 & 0 & 0.6 \\ 0.6 & -0.6 & 0 \end{pmatrix},$$

which forms a 3-cycle and yields an interior maximal lottery. For mixture weight $\alpha \in [0, 1]$, define the pooled matrix $M(\alpha) := \alpha M^{(\mathrm{EN})} + (1 - \alpha) M^{(\mathrm{ES})}$. Figure 2 visualizes the resulting maximal lotteries: (a) a Condorcet-winner vertex in EN, (c) an interior point in ES, and (b) sensitivity of the maximal lottery to nearby $\alpha$ and $\alpha'$ even when the preference directions are unchanged.

Maximal lotteries satisfy several axiomatic properties that are useful for evaluation (Brandl et al., 2016):

- **Nonparametric.** The method depends only on pairwise margins and does not assume transitivity or a latent utility scale.

- **Existence and convexity.** A maximal lottery exists for every $M \in \mathbb{M}_m$, and the set $\mathrm{ML}(M)$ is convex.

- **Condorcet consistency.** If there exists a *Condorcet winner* $i$ such that $M_{ij} > 0$ for all $j \neq i$, then the unique maximal lottery is the point mass $p^\star = e_i$.

- **Clone invariance.** If an alternative is cloned with identical pairwise margins, the maximal lottery only splits probability among clones and leaves all other probabilities unchanged.

- **Population consistency.** If $p$ is maximal for $M^{(1)}$ and $M^{(2)}$, then $p$ is also maximal for any convex combination $\alpha M^{(1)} + (1 - \alpha) M^{(2)}$.

- **Efficient computation.** A maximal lottery can be computed in polynomial time via linear programming.

## 4. Robust Lotteries

Section 3 treats evaluation as a single game built from pooled comparisons across, for example, subpopulations. Here, we drop the pooling assumption. As Section 3 illustrates, the maximal lottery can change sharply under small shifts in how subpopulations are weighted, motivating an explicit worst-case objective over a family of plausible margin matrices.

**Definition 4** (Ambiguity Set)**.** Let $\mathfrak{M}$ be the class of non-empty, compact, convex subsets of $\mathbb{M}_m$. An ambiguity set $\mathcal{M}$ is an element of $\mathfrak{M}$.[1]

The ambiguity set captures a range of preferences we wish to represent. For example, in our motivating example, it captures the diversity of preferences inherent across different user subpopulations. More generally, it can capture other notions such as statistical uncertainty about a true margin matrix (if it exists) or variation across evaluation tasks.

We impose three requirements on robust lotteries. First, they must reduce to maximal lotteries in the absence of uncertainty. Second, if an analyst is unsure which of two ambiguity sets is correct, they should evaluate a lottery by its worst guarantee across the two sets. Lastly, a lottery's score must vary continuously with the ambiguity set.

We now define robust lotteries. We show in Appendix A that the three requirements uniquely pin down this definition.

**Definition 5** (Robust Lotteries)**.** Given an ambiguity set $\mathcal{M} \in \mathfrak{M}$, define the robust value of a lottery $p \in \Delta(A)$ by

$$V(p, \mathcal{M}) := \min_{M \in \mathcal{M}} \min_{q \in \Delta(A)} p^\top M q.$$

A *robust lottery* is any maximizer of this value:

$$p^\star \in \arg\max_{p \in \Delta(A)} V(p, \mathcal{M}).$$

---

[1] We assume convexity for ease of exposition. Although our motivating example involves a finite set of distinct subpopulations, this will turn out to be equivalent to using their convex hull.

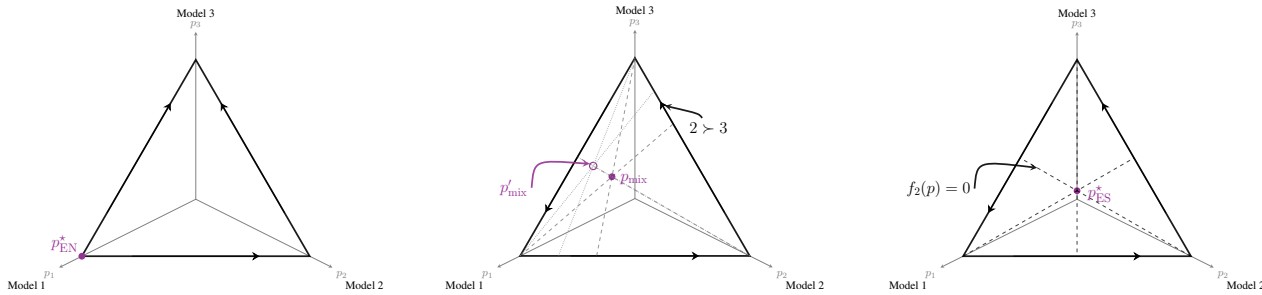

*Figure 2.* **Maximal lotteries are sensitive to population shifts.** Each simplex point is a lottery $p = (p_1, p_2, p_3)$ over models $\{1, 2, 3\}$ with vertices being deterministic choices. Edge arrows indicate the direction of preference. In each panel, the dashed lines are zero-margin boundaries $f_j(p) = p^\top M e_j = 0$, i.e., mixtures that tie pure opponent $j$ under the stratum margin matrix $M$. **Left:** English stratum has a Condorcet winner, so the maximal lottery collapses to the winner ($p_{\text{EN}}^\star$ at a vertex). **Right:** Spanish stratum exhibits a 3-cycle, so $p_{\text{ES}}^\star$ lies in the interior, balancing the worst-case opponent. **Middle:** two nearby population mixtures (weights $\alpha$ and $\alpha'$ over strata) induce two nearby matrices and thus two sets of zero-margin lines; their intersections give $p_{\text{mix}}^\star$ and $p_{\text{mix}}^\star{}'$, illustrating that even with the same majority directions, small shifts in mixture weights can move the maximal lottery. This sensitivity motivates robust maximal lotteries, which optimize a worst-case guarantee over a set of plausible population mixtures.

Let $\text{RL}(\mathcal{M})$ denote the set of robust lotteries for $\mathcal{M}$. We call $v^\star(\mathcal{M}) := \max_p V(p, \mathcal{M})$ the *robust game value*.

**Definition 6** (Robust Bipartisan Set). The *robust bipartisan set* is the set of alternatives that receive positive probability in at least one robust lottery:

$$\text{RBP}(\mathcal{M}) := \bigcup_{p \in \text{RL}(\mathcal{M})} \text{supp}(p).$$

### 4.1. Properties of Robust Lotteries

Robust lotteries extend the maximal lottery rule to ambiguity sets and satisfy many corresponding axiomatic properties. We first list basic properties of robust lotteries:

- **Nonparametric.** The method depends only on pairwise margins of the groups and does not assume transitivity or a latent utility scale.

- **Existence and convexity.** The set $\text{RL}(\mathcal{M})$ is nonempty and convex.

- **Neutrality.** The robust lottery is invariant under permutations of alternatives and groups, up to the corresponding relabeling.

- **Monotonicity.** Enlarging the ambiguity set cannot increase the robust game value.

In addition to these properties, we show that robust lotteries also satisfy the following analogs to the properties of maximal lotteries: **robust Condorcet consistency** (Theorem 1), **weak-clone invariance** (Theorem 3), and a **robust game value** guarantee under mixtures of ambiguity sets (Theorem 4). We relegate proofs of the basic properties and the following results to Appendix A.

**Definition 7** (Robust Condorcet Winner). An alternative $i^\star \in A$ is a robust Condorcet winner (RCW) if $M_{i^\star j} \geq 0$ for all $j \in A$ and all $M \in \mathcal{M}$. It is a strict RCW if, additionally, for every $j \neq i^\star$ there exists $\tilde{M} \in \mathcal{M}$ such that $\tilde{M}_{i^\star j} > 0$.

**Theorem 1** (Robust Condorcet Consistency). *If $i^\star$ is an RCW, then the point mass $e_{i^\star}$ is a robust lottery. If $i^\star$ is a strict RCW, then $e_{i^\star}$ is the unique robust lottery.*

To complement robust Condorcet consistency, we establish a robust dominance property: any alternative that is uniformly dominated throughout the ambiguity set is excluded from the support of every robust lottery.

**Theorem 2** (Robust Dominance). *Suppose there exist distinct alternatives $x, y \in A$ such that for every $M \in \mathcal{M}$ and every $j \in A$,*

$$M_{xj} > M_{yj}.$$

*Then every robust lottery $p \in \text{RL}(\mathcal{M})$ satisfies $p_y = 0$.*

Another important desideratum in social choice is *clone invariance*, which requires that introducing redundant alternatives does not distort the outcome among the original alternatives. This property is especially salient in language model evaluation, where it is easy to release multiple closely related checkpoints or minor variants and thereby crowd a leaderboard. Our result shows that robust lotteries are invariant to adding weaker clones of a model, defined as models that are indistinguishable from or strictly inferior to an existing model.[2]

**Definition 8** (Adding weaker clones). Fix $i \in A$ and add $\ell \geq 1$ variants $C = \{i^{(1)}, \ldots, i^{(\ell)}\}$. Consider the new set of

---

[2]Standard clone invariance typically considers exact clones: alternatives that perform identically against all non-clone opponents. Our formulation is strictly stronger, ensuring robustness even against inferior variants like legacy model checkpoints.

alternatives $\tilde{A} := A \cup C$. Given $M \in \mathbb{M}_{|A|}$, a *weak-clone expansion* of $M$ is any $\tilde{M} \in \mathbb{M}_{|\tilde{A}|}$ such that

$$
\begin{aligned}
\tilde{M}_{ab} &= M_{ab} && \forall a, b \in A, \\
\tilde{M}_{cx} &\leq \tilde{M}_{ix} && \forall c \in C, \, \forall x \in \tilde{A}.
\end{aligned}
$$

Equivalently, each added variant $c$ is pointwise no better than $i$ against every opponent. $\tilde{\mathcal{M}}$ is a weak-clone expansion of an ambiguity set $\mathcal{M}$ if

$$
\tilde{\mathcal{M}} := \{\tilde{M} \in \mathbb{M}_{|\tilde{A}|} : \exists M \in \mathcal{M}
$$

$$
\text{s.t. } \tilde{M} \text{ is a weak-clone expansion of } M\}.
$$

For $\tilde{p} \in \Delta(\tilde{A})$, define its projection onto $A$ by

$$
p_j = \tilde{p}_j \quad (j \in A \setminus \{i\}), \qquad p_i = \tilde{p}_i + \sum_{c \in C} \tilde{p}_c.
$$

**Theorem 3** (Weak-Clone Invariance). *If $\tilde{\mathcal{M}}$ is a weak-clone expansion of $\mathcal{M}$, then, for every $\tilde{p} \in \mathrm{RL}(\tilde{\mathcal{M}})$, its projected lottery $p$ is a robust lottery for $\mathcal{M}$. Consequently, the robust-lottery weight on any non-clone $j \in A \setminus \{i\}$ (and hence $j$'s membership in the robust bipartisan set) is unchanged by cloning. Moreover, if $i$ appears in the robust bipartisan set before cloning, it still appears after cloning.*

*Proof sketch.* Fix $\tilde{p} \in \mathrm{RL}(\tilde{\mathcal{M}})$ and let $\bar{p} \in \Delta(\tilde{A})$ be obtained by moving all probability mass that $\tilde{p}$ assigns to clones onto $i$. The weak-clone condition $\tilde{M}_{cx} \leq \tilde{M}_{ix}$ implies that, against any $\tilde{q}$, replacing a clone by $i$ can only increase the payoff $\tilde{p}^\top \tilde{M} \tilde{q}$. Hence $\bar{p}$ is also optimal for $\tilde{\mathcal{M}}$.

Let $p$ be the projected lottery on $A$ induced by $\bar{p}$ (equivalently by $\tilde{p}$). Comparing the inner minimizations shows that, for each $M \in \mathcal{M}$ and its expansion $\tilde{M}$, $\min_{\tilde{q}} \bar{p}^\top \tilde{M} \tilde{q} = \min_q p^\top M q$, so $V(\bar{p}, \tilde{\mathcal{M}}) = V(p, \mathcal{M})$. Since $\bar{p}$ is optimal for $\tilde{\mathcal{M}}$, this forces $p$ to be optimal for $\mathcal{M}$, proving the first claim. The invariance of weights for $j \in A \setminus \{i\}$ is immediate from the definition of the projection.

Finally, if $i$ is in the robust bipartisan set for $\mathcal{M}$, we can lift an optimal robust lottery on $A$ to $\tilde{A}$ by assigning zero mass to clones. The same comparison shows it remains optimal for $\tilde{\mathcal{M}}$, so $i$ remains in the bipartisan set after cloning. $\square$

Lastly, we discuss how *population consistency* relates to our setting. In the classical framework, maximality is characterized by linear inequalities in the margin matrix, and these constraints are preserved under convex combinations. Equivalently, if a lottery is maximal for each population, it remains maximal for their mixture. Our setup is qualitatively different. Robust lotteries are defined by a minimax criterion over an ambiguity set of margin matrices. The mixed set $\mathcal{M}_\lambda$ contains matrices formed by combining different elements of $\mathcal{M}_1$ and $\mathcal{M}_2$, and the identity of the worst-case matrix (which determines the minimax optimum) can therefore change under mixing. Accordingly, population consistency is not the appropriate invariance notion for minimax choice over ambiguity sets. Nevertheless, the robust game value behaves well under aggregation at the level of guarantees. In fact, it is lower bounded by the mixture of each game value; see Theorem 4.

**Definition 9** (Mixture of Ambiguity Sets). Let $\mathcal{M}_1, \mathcal{M}_2$ be two ambiguity sets of margin matrices. For $\lambda \in [0, 1]$, their population mixture is the Minkowski convex combination:

$$
\begin{aligned}
\mathcal{M}_\lambda &:= \lambda \mathcal{M}_1 \oplus (1 - \lambda) \mathcal{M}_2 \\
&= \{\lambda M_1 + (1 - \lambda) M_2 : M_1 \in \mathcal{M}_1, M_2 \in \mathcal{M}_2\}.
\end{aligned}
$$

**Definition 10** (Population Consistency). A social choice rule $F$ mapping ambiguity sets to sets of lotteries is *population-consistent* if, for any $\mathcal{M}_1, \mathcal{M}_2$ and any $\lambda \in (0, 1)$:

$$
p \in F(\mathcal{M}_1) \cap F(\mathcal{M}_2) \implies p \in F(\mathcal{M}_\lambda).
$$

**Theorem 4.** *Robust lotteries are not population-consistent for $m \geq 3$. However, when there exists a lottery $p$ that is a robust lottery for both $\mathcal{M}_1$ and $\mathcal{M}_2$, the robust game value achieved is stable under mixing populations,*

$$
v^\star(\mathcal{M}_\lambda) \geq \lambda v^\star(\mathcal{M}_1) + (1 - \lambda) v^\star(\mathcal{M}_2).
$$

There is a special case when robust lotteries are population-consistent. This happens when all margin matrices share a common strict RCW.

**Theorem 5** (Population consistency under a common strict RCW). *If there exists $i^\star \in A$ that is an RCW for both $\mathcal{M}_1$ and $\mathcal{M}_2$, then $i^\star$ is an RCW for $\mathcal{M}_\lambda$. Consequently, $e_{i^\star} \in \mathrm{RL}(\mathcal{M}_\lambda)$. If $i^\star$ is a strict RCW for $\mathcal{M}_1$ and $\mathcal{M}_2$, then robust lotteries are population-consistent.*

### 4.2. Computing Robust Lotteries

Our robust-lottery objective is a minimax problem over an ambiguity set of margin matrices. In full generality, this set can be arbitrary, and solving the corresponding minimax program can be computationally prohibitive. This subsection isolates a specific instantiation of uncertainty over the *subpopulation mixture weights* that admits efficient optimization and finite-sample guarantees.

Consider $K$ subpopulations, each inducing a margin matrix $M^{(k)} \in \mathbb{M}_m$. Any distribution over subpopulations corresponds to a mixture:

$$
M(w) := \sum_{k=1}^{K} w_k M^{(k)}, \qquad w \in \Delta(K).
$$

To model uncertainty in the mixture weights, we allow $w$ to vary over a convex and non-empty set $\mathcal{W} \subseteq \Delta(K)$, which gives the ambiguity set:

$$\mathcal{M} := \{M(w) : w \in \mathcal{W}\}.$$

Fix a reference mixture $w_0 \in \Delta(K)$. For a radius $\rho \in [0, 1]$, we take $\mathcal{W}$ to be the $\ell_1$ ball:

$$\mathcal{W}(w_0, \rho) := \left\{w \in \Delta(K) : \tfrac{1}{2}\|w - w_0\|_1 \leq \rho\right\}.$$

This choice provides a continuous interpolation between standard maximal lottery over the aggregate matrix ($\rho = 0$) and worst-case robustness over all mixtures ($\rho = 1$). The $\ell_1$ ball has a clean distributional interpretation: $\mathcal{W}(w_0, \rho)$ consists of the set of distributions over groups that are within a total variation distance $\rho$ of the original distribution $w_0$. Intuitively, this corresponds to reallocating up to a $\rho$-fraction of the mixture weight from some groups to others. This models shifts in prompt or annotator mix without requiring a parametric shift model.

**Definition 11** (Distributionally Robust Lottery). Consider the robust value function defined over $\mathcal{W}(w_0, \rho)$:

$$V(p, \mathcal{W}(w_0, \rho)) = \min_{w \in \mathcal{W}(w_0, \rho)} \min_{q \in \Delta(A)} p^\top M(w)\, q.$$

A distributionally robust lottery (DRL) is any distribution $p \in \Delta(A)$ that maximizes the worst-case game value over admissible mixtures:

$$p^\star \in \arg\max_{p \in \Delta(A)} V(p, \mathcal{W}(w_0, \rho)). \tag{2}$$

We refer to the robust game value as $v^\star(\mathcal{W}(w_0, \rho))$.

From a computational standpoint, uncertainty in $\ell_1$ distance leads to a linear program through standard duality, which yields an $O(mK)$-size formulation in Theorem 6.

**Theorem 6** (LP formulation for DRL). *The distributionally robust lottery problem in Equation (2) can be written as a linear program of size $O(mK)$:*

$$\max_{p,t,\mu,\lambda,\gamma} \quad t$$

$$\text{s.t.} \quad \sum_{i \in A} p_i = 1, \quad p_i \geq 0 \;\; \forall i,$$
$$t \leq \mu_a - 2\rho\,\lambda_a + \sum_{k=1}^{K} w_{0,k}\gamma_{a,k} \quad \forall a \in A,$$
$$\mu_a + \gamma_{a,k} \leq p^\top M^{(k)} e_a \quad \forall a \in A, \forall k \in [K],$$
$$-\lambda_a \leq \gamma_{a,k} \leq \lambda_a \quad \forall a \in A, \forall k \in [K],$$
$$\lambda_a \geq 0 \quad \forall a \in A.$$

An immediate benefit of the LP formulation is that it accommodates additional linear constraints on the lottery, e.g., inference budget, latency, or minimum performance on an external benchmark. For example, we exploit this flexibility in Figure 1 to compute the cost–performance frontier for the robust lottery with $\rho = 1$.

Moreover, there exists a *sparse* lottery, with support logarithmic in both the number of models and groups, that is $\varepsilon$-close to optimal. This supports the empirical pattern that the robust bipartisan set is typically small compared to the set of alternatives.

**Theorem 7** (Sparse $\varepsilon$-optimal DRL). *Fix $w_0 \in \Delta(K)$ and $\varepsilon \in (0, 1]$. Consider any $\rho$ satisfying $\rho \leq \min\{\min_{k \in [K]} w_{0,k},\ 1 - \max_{k \in [K]} w_{0,k}\}$. Then there exists $p \in \Delta(A)$ with*

$$|\mathrm{supp}(p)| \leq \max\left\{\left(\frac{8}{\varepsilon^2}\log(4m)\right), \left(\frac{32\rho^2}{\varepsilon^2}\log(8mK)\right)\right\}$$

*such that $V(p, \mathcal{W}(w_0, \rho)) \geq v^\star(\mathcal{W}(w_0, \rho)) - \varepsilon$.*

**Choosing $\rho$ from data.** So far, $\rho$ controls robustness to shifts in the (unknown) subpopulation weights. Even without explicit distribution shift, the DRL can also stabilize the solution against finite sample error when one only has access to an empirical estimate of weights $\hat{w}$. When $\rho = 0$, the DRL reduces to the maximal lottery for the single empirical pooled matrix $M(\hat{w})$. Since $\hat{w}$ is random, this can yield a lottery that performs poorly under the *true population* mixture $M(w^\star)$. The next result gives a data-dependent radius $\rho(n, \delta)$ that guarantees small regret under $w^\star$.

Formally, fix group-specific margin matrices $M^{(1)}, \ldots, M^{(K)} \in \mathbb{M}_m$, and let $w^\star \in \Delta(K)$ denote the (unknown) population mixture that generates the preference dataset. We observe i.i.d. group labels $Z_1, \ldots, Z_n \sim w^\star$ and form $\hat{w} \in \Delta(K)$,

$$\hat{w}_k := \frac{1}{n}\sum_{t=1}^{n} \mathbf{1}\{Z_t = k\}, \qquad k \in [K].$$

The value for a pair $(p, w)$ is

$$v(p, w) := \min_{q \in \Delta(A)} p^\top M(w)\, q.$$

Let $p^\star$ be the maximal lottery for $M(w^\star)$, then $v(p^\star, w^\star) = 0$. Given $\rho \geq 0$, we compute the empirical DRL

$$\hat{p} \in \arg\max_{p \in \Delta(A)} \min_{w \in \mathcal{W}(\hat{w}, \rho)} v(p, w). \tag{3}$$

We measure performance under the true $w^\star$ via the regret

$$\mathrm{Regret}(w^\star) := v(p^\star, w^\star) - v(\hat{p}, w^\star) = -v(\hat{p}, w^\star).$$

**Theorem 8** (Regret bound for DRL). *Fix $\delta \in (0, 1)$. Set*

$$\rho = \rho(n, \delta) := \min\left\{1, \sqrt{\frac{K}{n}} + \sqrt{\frac{2}{n}\log\left(\frac{2}{\delta}\right)}\right\},$$

*and compute the empirical DRL in Equation (3). Then, with probability at least $1 - \delta$, we have that $w^\star \in \mathcal{W}(\hat{w}, \rho)$ and*

$$\text{Regret}(w^\star) \leq 4\sqrt{\frac{K}{n}} + 4\sqrt{\frac{2}{n}\log\left(\frac{2}{\delta}\right)}.$$

This regret bound gives a theoretically principled way to select $\rho$ when weights are estimated from finite samples. We later demonstrate in experiments the consequences of using various $\rho$ in practice.

## 5. Experiments

Having established the theoretical properties of robust lotteries, we now evaluate robust lotteries on several real language model leaderboards.

**Leaderboard 1: LMArena (Chiang et al., 2024).** We consider the most recent publicly released datasets from LMArena as of time of submission. We categorize annotators into groups based on the language of the prompts they themselves generated, which is the only identifier provided in the dataset.

**Leaderboard 2: HUMAINE (Prolific AI, 2025).** HUMAINE is a collection of pairwise preferences across diverse demographic groups and conversation contexts. We categorize annotators based on their expressed ethnic group.

We provide details on the experimental setup and results on two additional leaderboards (SearchArena and Open LLM) in Appendix B.

### 5.1. Results

**Preferences vary sharply across groups.** Different groups often prefer different models, so a single pooled ranking or maximal lottery can obscure heterogeneity. To quantify this, we compute the probability of a *preference reversal* across groups for each model pair $(i, j)$:

$$\Pr\left[\text{sign}\big(M_{ij}^{(k)}\big) \neq \text{sign}\big(M_{ij}^{(k')}\big)\right], \ \ k, k' \overset{\text{i.i.d.}}{\sim} \hat{w}, \ \ k \neq k',$$

where $\hat{w}$ is the empirical group-frequency distribution in the dataset. In LMArena, the five model pairs with the largest disagreement have reversal rates of 42%–47%, typically driven by English versus non-English prompt categories (e.g., Chinese, Polish, Russian); see Figure B.2 in Appendix B. For example, Gemini 2.0 Flash is substantially more preferred in non-English groups than the overall stronger model Opus 4 Thinking.

**Robust lotteries improve worst-group test performance (Figure 3).** We compute robust lotteries for a range of radii $\rho$. Across leaderboards, increasing $\rho$ improves the worst-group win rate on held-out votes, with a modest decrease

for the best-performing groups. Moreover, on LMArena, we show that robust lotteries outperform every single-model baseline (including the top-ranked model according to BT) on every subgroup for a moderate choice of $\rho$; see Table 1.

**Robust lotteries identify a set of frontier models (Figure 4).** Improving the worst-group win rate requires placing additional weight on models that are comparatively stronger in the lower-performing groups. As $\rho$ increases, the learned mixture expands beyond the top aggregate model and assigns mass to a small set of complementary models.

## 6. Discussion

AI evaluation inevitably relies on a sequence of reductions, since complex preference data – often collected from thousands of users – is mapped onto a single human-understandable ranking or set of "good" models. However, if we ignore the inherent heterogeneity in preferences, our findings show that we risk rewarding models that fail on specific tasks or user populations when deployed in the wild.

Robust lotteries are a step towards addressing this issue, offering an evaluation method that captures variations across users and tasks. While they do not directly provide a "leaderboard" – see the work of Lanctot et al. (2025) that extends lotteries to a ranking – lotteries offer practitioners the ability to navigate tradeoffs in the model ecosystem and find the best set of models under real-world constraints. For example, robust lotteries can be used to construct a benchmark of frontier models: as new models are released, the benchmark shows whether they enter the frontier and at what cost level. Unlike a single ranking, this can capture multiple models with complementary strengths. Moreover, robust lotteries offer a principled method for model selection. Since robust lotteries directly optimize for downstream performance, one can restrict attention to the frontier models at the relevant budget, rather than test every available model, and then select among them using some application-specific criteria. Ultimately, prioritizing pluralistic approaches that consider worst-case performance, like robust lotteries, is a necessary step for ensuring that AI systems work for everyone.

*Table 1.* **Robust lotteries can provide stronger win rate guarantees than any algorithm that picks a single winner, including BT.** We compare the robust lottery against the maximal lottery and the strongest single-model baselines on LMArena. The robust lottery attains the highest win rate in every subgroup and the best overall performance on held-out votes, indicating stronger empirical robustness across preference populations.

| Model / Mixture | English | Polish | Chinese | Russian | Overall |
|---|---|---|---|---|---|
| Deepseek R1 | $31.25 \pm 0.70$ | $24.76 \pm 0.57$ | $26.13 \pm 0.52$ | $26.17 \pm 0.63$ | $36.92 \pm 0.50$ |
| Gemini-2.5 Pro | $40.97 \pm 0.66$ | $31.13 \pm 0.85$ | $29.48 \pm 0.72$ | $25.76 \pm 0.54$ | $44.22 \pm 0.50$ |
| Llama 4 Maverick | $24.70 \pm 0.70$ | $25.53 \pm 0.68$ | $29.75 \pm 0.77$ | $25.80 \pm 0.58$ | $32.55 \pm 0.56$ |
| Maximal Lottery | $40.78 \pm 0.66$ | $32.69 \pm 0.83$ | $32.06 \pm 0.72$ | $28.99 \pm 0.68$ | $44.06 \pm 0.48$ |
| Robust Lottery ($\rho = 0.4$) | $\mathbf{41.89} \pm 0.44$ | $\mathbf{38.01} \pm 0.56$ | $\mathbf{37.09} \pm 0.54$ | $\mathbf{35.71} \pm 0.55$ | $\mathbf{44.24} \pm 0.32$ |

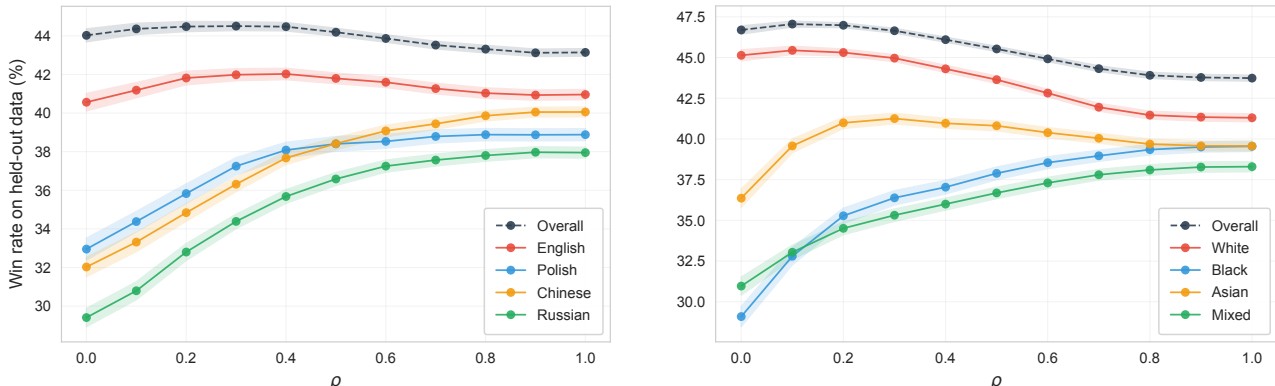

*Figure 3.* **Robust lotteries improve win rate guarantees across subpopulations.** We compute robust lotteries for varying radius values $\rho$ and evaluate each lottery on held-out votes (20% split). We show bootstrap means of win rates achieved on the overall population and each subgroup with standard errors (200 samples). As $\rho$ increases, robust lotteries improve the win rate guarantees for the lowest-performing groups with a modest decrease for the highest-performing groups, illustrating a robustness–accuracy trade-off. **Left:** LMArena, with groups defined by prompt language. **Right:** HUMAINE, with groups defined by annotator's ethnic group.

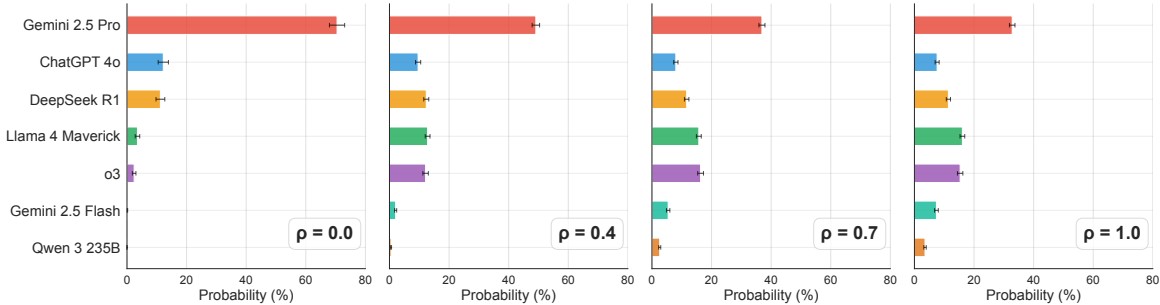

*Figure 4.* **Robust lotteries diversify the lottery to handle preference tradeoffs among subpopulations.** We present the estimated probability assigned to each model with its standard error from LMArena. At $\rho = 0$, the lottery concentrates on the top aggregate performer (Gemini 2.5 Pro). As $\rho$ increases, probability mass shifts toward additional strong models (e.g., o3 and Llama 4 Maverick), reflecting the need to hedge to improve win rate guarantees as seen in Figure 3.

## Acknowledgments

This work was partially supported by the National Science Foundation under grant IIS-2229881, CIF-2231707, and CIF-2312667; by the Office of Naval Research under grants N00014-24-1-2704 and N00014-25-1-2153; by grants from the Cooperative AI Foundation, the Foresight Institute, and Coefficient Giving; and a faculty award by JPMorganChase.

## Impact Statement

The techniques presented in this work seek to advance practices around AI evaluation. This has potential impacts on public leaderboards of AI models (the primary focus of our experiments), which could impact both developers and consumers of AI systems. We discuss both types of stakeholders in more detail below. Overall, we are optimistic that the proposed robust lotteries have the potential to improve

the health of the AI development ecosystem.

From the standpoint of consumers, models selected by our robust lotteries are guaranteed to be useful across a diverse user population with heterogeneous preferences. The robust lotteries also protect against loss of validity due to distribution shift over time. Taken together, robust lotteries could help better inform consumer choices. From the standpoint of model developers, robust lotteries would surface models that perform well on subpopulations of users and prompts, rather than only those models that perform well overall. We believe this would help deter a monoculture of very similar AI systems on the market (Wu et al., 2025).

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

# A. Additional Proofs

In this section, we provide additional results and proofs.

**Definition 12** (Admissible Robust Score). A functional $V : \Delta(A) \times \mathfrak{M} \to \mathbb{R}$ is an admissible robust score if for every lottery $p \in \Delta(A)$ the following hold:

1. For all $M \in \mathbb{M}_m$, $V(p, \{M\}) = \min_{q \in \Delta(A)} p^\top M q$.

2. For all $\mathcal{M}_1, \mathcal{M}_2 \in \mathfrak{M}$,
$$V(p, \mathcal{M}_1 \vee \mathcal{M}_2) = \min\{V(p, \mathcal{M}_1), V(p, \mathcal{M}_2)\}$$
where $\mathcal{M}_1 \vee \mathcal{M}_2$ is the smallest convex ambiguity set that contains both $\mathcal{M}_1$ and $\mathcal{M}_2$.

3. The map $\mathcal{M} \mapsto V(p, \mathcal{M})$ is continuous on $\mathfrak{M}$ with respect to the Hausdorff metric induced by $\| \cdot \|_\infty$.

**Theorem 1** (Characterization of Robust Lotteries). *Let $V$ be an admissible robust score in the sense of Definition 12. Then for every lottery $p \in \Delta(A)$ and ambiguity set $\mathcal{M} \in \mathfrak{M}$,*
$$V(p, \mathcal{M}) = \min_{M \in \mathcal{M}} \min_{q \in \Delta(A)} p^\top M q.$$

*The social choice rule induced by $V$ is*
$$F(\mathcal{M}) := \arg \max_{p \in \Delta(A)} V(p, \mathcal{M}).$$

*Consequently, the induced correspondence is exactly the robust lottery. Moreover, this is the unique score-based rule satisfying Definition 12.*

*Proof of Theorem 1.* Fix $p \in \Delta(A)$ and consider $v(p, M) := \min_{q \in \Delta(A)} p^\top M q$. For any $M, M' \in \mathbb{M}_m$, we have that:
$$|v(p, M) - v(p, M')| = \left| \min_q p^\top M q - \min_q p^\top M' q \right| \leq \max_{q \in \Delta(A)} |p^\top (M - M') q| \leq \|M - M'\|_\infty,$$

since $p^\top (M - M') q$ is a convex combination of the entries of $M - M'$.

Assume there exists $V$ satisfying the three stated conditions. Let $\mathcal{M} \in \mathfrak{M}$. We claim that for every $p \in \Delta(A)$,
$$V(p, \mathcal{M}) = \min_{M \in \mathcal{M}} v(p, M). \tag{4}$$

To prove Equation (4), we begin with the case where $\mathcal{M}$ is the convex hull of finitely many matrices. Consider $M^{(1)}, \ldots, M^{(k)} \in \mathbb{M}_m$ and set $\mathcal{P} := \text{conv}\{M^{(1)}, \ldots, M^{(k)}\} \in \mathfrak{M}$. By definition of the operator $\vee$, we have $\mathcal{P} = \{M^{(1)}\} \vee \cdots \vee \{M^{(k)}\}$, so repeated use of property (2) yields
$$V(p, \mathcal{P}) = \min_{1 \leq i \leq k} V(p, \{M^{(i)}\}).$$

By property (1), $V(p, \{M^{(i)}\}) = v(p, M^{(i)})$, hence
$$V(p, \mathcal{P}) = \min_{1 \leq i \leq k} v(p, M^{(i)}). \tag{5}$$

Now consider an arbitrary $\mathcal{M} \in \mathfrak{M}$. For each $\varepsilon > 0$, choose a finite $\varepsilon$-net $\mathcal{N}_\varepsilon \subset \mathcal{M}$ in $\| \cdot \|_\infty$ and set $\mathcal{P}_\varepsilon := \text{conv}(\mathcal{N}_\varepsilon)$. Then $\mathcal{P}_\varepsilon \subseteq \mathcal{M}$, and every $M \in \mathcal{M}$ lies within $\varepsilon$ (in $\| \cdot \|_\infty$) of some element of $\mathcal{N}_\varepsilon \subseteq \mathcal{P}_\varepsilon$. It follows that the Hausdorff distance satisfies $d_H(\mathcal{P}_\varepsilon, \mathcal{M}) \leq \varepsilon$. By property (3),
$$V(p, \mathcal{M}) = \lim_{\varepsilon \to 0} V(p, \mathcal{P}_\varepsilon).$$

Using Equation (5) for $\mathcal{P}_\varepsilon = \text{conv}(\mathcal{N}_\varepsilon)$,
$$V(p, \mathcal{P}_\varepsilon) = \min_{M \in \mathcal{N}_\varepsilon} v(p, M).$$

We compare this minimum over the net to $\min_{M \in \mathcal{M}} v(p, M)$. Since $\mathcal{N}_\varepsilon \subseteq \mathcal{M}$,

$$\min_{M \in \mathcal{M}} v(p, M) \ \leq \ \min_{M \in \mathcal{N}_\varepsilon} v(p, M). \tag{6}$$

Conversely, compactness of $\mathcal{M}$ and continuity of $v(p, \cdot)$ imply the minimum is attained since we can choose $M^\star \in \arg\min_{M \in \mathcal{M}} v(p, M)$. Pick $\hat{M} \in \mathcal{N}_\varepsilon$ with $\|M^\star - \hat{M}\|_\infty \leq \varepsilon$. By the Lipschitz estimate above,

$$v(p, \hat{M}) \leq v(p, M^\star) + \varepsilon = \min_{M \in \mathcal{M}} v(p, M) + \varepsilon,$$

hence

$$\min_{M \in \mathcal{N}_\varepsilon} v(p, M) \ \leq \ \min_{M \in \mathcal{M}} v(p, M) + \varepsilon. \tag{7}$$

Combining Equation (6) and Equation (7) and taking $\varepsilon \to 0$ gives

$$\lim_{\varepsilon \to 0} \min_{M \in \mathcal{N}_\varepsilon} v(p, M) = \min_{M \in \mathcal{M}} v(p, M).$$

Therefore,

$$V(p, \mathcal{M}) = \lim_{\varepsilon \to 0} V(p, \mathcal{P}_\varepsilon) = \lim_{\varepsilon \to 0} \min_{M \in \mathcal{N}_\varepsilon} v(p, M) = \min_{M \in \mathcal{M}} v(p, M),$$

which proves Equation (4) and hence $F = \mathrm{RL}$.

For the converse direction, assume $F = \mathrm{RL}$ and define

$$V(p, \mathcal{M}) := \min_{M \in \mathcal{M}} \min_{q \in \Delta(A)} p^\top M q.$$

Property (1) is immediate. Property (2) holds because for fixed $p$ the map $M \mapsto v(p, M)$ is concave since it is the pointwise minimum of linear functionals in $M$. The minimum of $v(p, \cdot)$ over $\mathrm{conv}(\mathcal{M}_1 \cup \mathcal{M}_2)$ equals its minimum over $\mathcal{M}_1 \cup \mathcal{M}_2$, which is $\min\{\min_{\mathcal{M}_1} v(p, \cdot), \min_{\mathcal{M}_2} v(p, \cdot)\}$. Property (3) follows from the Lipschitz bound $|v(p, M) - v(p, M')| \leq \|M - M'\|_\infty$ and the fact that taking a minimum of a uniformly continuous function over a compact set is continuous with respect to the Hausdorff metric. By construction $F(\mathcal{M}) = \arg\max_p V(p, \mathcal{M})$, so the stated representation holds. $\qquad \square$

*Proof of basic properties of robust lotteries.* Recall $V(p, \mathcal{M}) := \min_{M \in \mathcal{M}} \min_{q \in \Delta(A)} p^\top M q$ and $v^\star(\mathcal{M}) := \max_{p \in \Delta(A)} V(p, \mathcal{M})$, with $\mathrm{RL}(\mathcal{M}) = \arg\max_{p \in \Delta(A)} V(p, \mathcal{M})$.

**Existence and convexity.** For each fixed $(M, q)$, the map $p \mapsto p^\top M q$ is continuous and linear. Hence $p \mapsto V(p, \mathcal{M})$, being the pointwise infimum of a family of continuous linear functions, is upper semi-continuous and concave on the compact set $\Delta(A)$. Therefore $V(\cdot, \mathcal{M})$ attains a maximum on $\Delta(A)$, so $\mathrm{RL}(\mathcal{M})$ is non-empty.

To see convexity, let $p_1, p_2 \in \mathrm{RL}(\mathcal{M})$ and let $v^\star = v^\star(\mathcal{M})$. For any $\lambda \in [0, 1]$ set $p_\lambda = \lambda p_1 + (1 - \lambda)p_2$. By concavity of $V(\cdot, \mathcal{M})$,

$$V(p_\lambda, \mathcal{M}) \ \geq \ \lambda V(p_1, \mathcal{M}) + (1 - \lambda)V(p_2, \mathcal{M}) = \lambda v^\star + (1 - \lambda)v^\star = v^\star.$$

Since $v^\star$ is the maximum value, $V(p_\lambda, \mathcal{M}) = v^\star$ and thus $p_\lambda \in \mathrm{RL}(\mathcal{M})$. Hence $\mathrm{RL}(\mathcal{M})$ is convex.

**Neutrality.** It is immediate that robust lotteries are invariant under relabeling groups. We prove invariance with respect to relabeling alternatives. Let $\sigma$ be a permutation with associated permutation matrix $P_\sigma$, and define $\mathcal{M}^\sigma = \{P_\sigma M P_\sigma^\top : M \in \mathcal{M}\}$. For all $p, q \in \Delta(A)$ and $M \in \mathcal{M}$,

$$(P_\sigma p)^\top (P_\sigma M P_\sigma^\top)(P_\sigma q) = p^\top M q.$$

Since $\Delta(A)$ is invariant under $P_\sigma$, it follows that $V(P_\sigma p, \mathcal{M}^\sigma) = V(p, \mathcal{M})$ for all $p$, and therefore $\mathrm{RL}(\mathcal{M}^\sigma) = P_\sigma \mathrm{RL}(\mathcal{M})$.

**Monotonicity.** If $\mathcal{M}_1 \subseteq \mathcal{M}_2$, then for any $p \in \Delta(A)$,

$$V(p, \mathcal{M}_2) = \min_{M \in \mathcal{M}_2} \min_{q \in \Delta(A)} p^\top M q \ \leq \ \min_{M \in \mathcal{M}_1} \min_{q \in \Delta(A)} p^\top M q = V(p, \mathcal{M}_1).$$

Taking $\max_{p \in \Delta(A)}$ on both sides yields $v^\star(\mathcal{M}_2) \leq v^\star(\mathcal{M}_1)$. $\qquad \square$

*Proof of Theorem 1.* Since $q \mapsto p^\top M q$ is linear and $\Delta(A) = \mathrm{conv}\{e_j : j \in A\}$, we have

$$\min_{q \in \Delta(A)} p^\top M q = \min_{j \in A} p^\top M e_j. \tag{8}$$

Also, for any $p \in \Delta(A)$ and any skew-symmetric $M$, we have $p^\top M p = 0$, hence

$$\min_{q \in \Delta(A)} p^\top M q \le p^\top M p = 0$$

Hence, $V(p, \mathcal{M}) \le 0$. Assume $i^\star$ is a robust Condorcet winner, i.e. $M_{i^\star j} \ge 0$ for all $j \in A$ and all $M \in \mathcal{M}$. Fix $M \in \mathcal{M}$. By Equation (8),

$$\min_{q \in \Delta(A)} e_{i^\star}^\top M q = \min_{j \in A} e_{i^\star}^\top M e_j = \min_{j \in A} M_{i^\star j}.$$

By the RCW condition, $M_{i^\star j} \ge 0$ for all $j$, and $M_{i^\star i^\star} = 0$, so $\min_j M_{i^\star j} = 0$. Therefore, we have that

$$V(e_{i^\star}, \mathcal{M}) = \min_{M \in \mathcal{M}} 0 = 0.$$

Since $V(p, \mathcal{M}) \le 0$ for all $p$, it follows that $e_{i^\star}$ maximizes $V(\cdot, \mathcal{M})$, i.e. $e_{i^\star} \in \mathrm{RL}(\mathcal{M})$.

As for uniqueness, $V(e_{i^\star}, \mathcal{M}) = 0$, hence $v^\star(\mathcal{M}) = 0$. Let $p \in \Delta(A)$ with $p \ne e_{i^\star}$. Then there exists $j \ne i^\star$ with $p_j > 0$. Choose $M^{(j)} \in \mathcal{M}$ such that $M_{i^\star j}^{(j)} > 0$. Take the pure action $q = e_{i^\star}$. Then

$$p^\top M^{(j)} q = p^\top M^{(j)} e_{i^\star} = \sum_{a \in A} p_a M_{a i^\star}^{(j)}.$$

By skew-symmetry, $M_{a i^\star}^{(j)} = -M_{i^\star a}^{(j)} \le 0$ for all $a$ (since $i^\star$ is an RCW), and moreover for $a = j$ we have

$$M_{j i^\star}^{(j)} = -M_{i^\star j}^{(j)} < 0.$$

Since $p_j > 0$, it follows that

$$p^\top M^{(j)} e_{i^\star} < 0.$$

Therefore

$$V(p, \mathcal{M}) = \min_{M \in \mathcal{M}} \min_{q \in \Delta(A)} p^\top M q \le \min_{q \in \Delta(A)} p^\top M^{(j)} q \le p^\top M^{(j)} e_{i^\star} < 0.$$

Thus every $p \ne e_{i^\star}$ achieves strictly negative robust value, while $e_{i^\star}$ achieves $0 = v^\star(\mathcal{M})$. Hence $\mathrm{RL}(\mathcal{M}) = \{e_{i^\star}\}$. $\quad\square$

*Proof of Theorem 2.* Let $p \in \Delta(A)$ with $p_y > 0$ and define $p' := p + p_y(e_x - e_y)$. Fix any $M \in \mathcal{M}$ and any $j \in A$. Using $e_x^\top M e_j = M_{xj}$,

$$p'^\top M e_j - p^\top M e_j = p_y(e_x - e_y)^\top M e_j = p_y(M_{xj} - M_{yj}) > 0.$$

Thus $p'^\top M e_j > p^\top M e_j$ for all $j$, hence

$$\min_{j \in A} p'^\top M e_j > \min_{j \in A} p^\top M e_j.$$

Taking $\min_{M \in \mathcal{M}}$ and using $\min_{q \in \Delta(A)} p^\top M q = \min_j p^\top M e_j$ gives $V(p', \mathcal{M}) > V(p, \mathcal{M})$.

Therefore, any $p \in \Delta(A)$ with $p_y > 0$ can be strictly improved (in robust value) by moving its mass on $y$ to $x$. In particular, no maximizer of $V(\cdot, \mathcal{M})$ can assign positive probability to $y$, so every robust lottery satisfies $p_y = 0$. $\quad\square$

*Proof of Theorem 3.* For $\tilde{r} \in \Delta(\tilde{A})$, write $\Pi \tilde{r} \in \Delta(A)$ for its projected lottery as in Definition 8. Let $\tilde{p} \in \mathrm{RL}(\tilde{\mathcal{M}})$ and define $\bar{p} \in \Delta(\tilde{A})$ by pushing all clone mass onto $i$:

$$\bar{p}_a = \tilde{p}_a \ (a \ne i), \qquad \bar{p}_i = \tilde{p}_i + \sum_{c \in C} \tilde{p}_c, \qquad \bar{p}_c = 0 \ (c \in C).$$

By construction, $\Pi \bar{p} = \Pi \tilde{p}$.

Fix $M \in \mathcal{M}$ and let $\tilde{M} = f(M) \in \tilde{\mathcal{M}}$ be its weak-clone expansion. For any $\tilde{q} \in \Delta(\tilde{A})$,

$$\bar{p}^\top \tilde{M} \tilde{q} - \tilde{p}^\top \tilde{M} \tilde{q} = \sum_{c \in C} \tilde{p}_c \left( \tilde{M}_{i\cdot} \tilde{q} - \tilde{M}_{c\cdot} \tilde{q} \right)$$
$$= \sum_{c \in C} \tilde{p}_c \sum_{x \in \tilde{A}} (\tilde{M}_{ix} - \tilde{M}_{cx}) \tilde{q}_x \ \geq \ 0,$$

since $\tilde{M}_{cx} \leq \tilde{M}_{ix}$ for all $c \in C$ and $x \in \tilde{A}$. Taking $\min_{\tilde{q} \in \Delta(\tilde{A})}$ and then $\min_{\tilde{M} \in \tilde{\mathcal{M}}}$ yields $V(\bar{p}, \tilde{\mathcal{M}}) \geq V(\tilde{p}, \tilde{\mathcal{M}})$. Because $\tilde{p}$ maximizes $V(\cdot, \tilde{\mathcal{M}})$, it follows that $\bar{p} \in \mathrm{RL}(\tilde{\mathcal{M}})$.

Set $p := \Pi\bar{p} \in \Delta(A)$. We show $p \in \mathrm{RL}(\mathcal{M})$. Fix $M \in \mathcal{M}$ and $\tilde{M} = f(M)$. For any $\tilde{q} \in \Delta(\tilde{A})$, let $q := \Pi\tilde{q} \in \Delta(A)$. Since $\bar{p}$ is supported on $A$,

$$\bar{p}^\top \tilde{M} \tilde{q} = \sum_{a \in A} p_a \sum_{x \in \tilde{A}} \tilde{M}_{ax} \tilde{q}_x = \sum_{a \in A} p_a \left( \sum_{b \in A} M_{ab} \tilde{q}_b + \sum_{c \in C} \tilde{M}_{ac} \tilde{q}_c \right).$$

For each $a \in A$ and $c \in C$, skew-symmetry gives $\tilde{M}_{ac} = -\tilde{M}_{ca}$, and the weak-clone condition with $x = a \in A$ gives $\tilde{M}_{ca} \leq \tilde{M}_{ia} = M_{ia}$. Hence

$$\tilde{M}_{ac} = -\tilde{M}_{ca} \ \geq \ -M_{ia} = M_{ai}.$$

Therefore,

$$\bar{p}^\top \tilde{M} \tilde{q} \geq \sum_{a \in A} p_a \left( \sum_{b \in A} M_{ab} \tilde{q}_b + \sum_{c \in C} M_{ai} \tilde{q}_c \right) = \sum_{a,b \in A} p_a M_{ab} q_b = p^\top M q.$$

Minimizing over $\tilde{q}$ gives

$$\min_{\tilde{q} \in \Delta(\tilde{A})} \bar{p}^\top \tilde{M} \tilde{q} \ \geq \ \min_{q \in \Delta(A)} p^\top M q.$$

Conversely, for any $q \in \Delta(A)$, define $\tilde{q} \in \Delta(\tilde{A})$ by $\tilde{q}|_A = q$ and $\tilde{q}_c = 0$ for all $c \in C$. Then $\Pi\tilde{q} = q$ and, since $\tilde{M}$ agrees with $M$ on $A \times A$, $\bar{p}^\top \tilde{M} \tilde{q} = p^\top M q$. Thus

$$\min_{\tilde{q} \in \Delta(\tilde{A})} \bar{p}^\top \tilde{M} \tilde{q} \ \leq \ \min_{q \in \Delta(A)} p^\top M q.$$

Combining yields, for each $M \in \mathcal{M}$,

$$\min_{\tilde{q} \in \Delta(\tilde{A})} \bar{p}^\top \tilde{M} \tilde{q} = \min_{q \in \Delta(A)} p^\top M q.$$

Minimizing over $M \in \mathcal{M}$ gives

$$V(\bar{p}, \tilde{\mathcal{M}}) = V(p, \mathcal{M}).$$

Now let $r \in \Delta(A)$ be arbitrary and lift it to $\tilde{r} \in \Delta(\tilde{A})$ by $\tilde{r}|_A = r$ and $\tilde{r}_c = 0$ for all $c \in C$. The same argument as above shows $V(\tilde{r}, \tilde{\mathcal{M}}) = V(r, \mathcal{M})$. Since $\bar{p} \in \mathrm{RL}(\tilde{\mathcal{M}})$, we have $V(\bar{p}, \tilde{\mathcal{M}}) \geq V(\tilde{r}, \tilde{\mathcal{M}})$, hence $V(p, \mathcal{M}) \geq V(r, \mathcal{M})$ for all $r$. Therefore $p \in \mathrm{RL}(\mathcal{M})$.

Finally, $\Pi\tilde{p} = \Pi\bar{p} = p$, so the projected lottery of $\tilde{p}$ is a robust lottery for $\mathcal{M}$, proving the first claim of the theorem. The second claim (weights and bipartisan membership for $j \neq i$) is immediate from the definition of the projection.

For the last claim, suppose $i$ appears in the robust bipartisan set for $\mathcal{M}$. Then there exists $p \in \mathrm{RL}(\mathcal{M})$ with $p_i > 0$. Let $\tilde{p} \in \Delta(\tilde{A})$ be its lift with $\tilde{p}|_A = p$ and $\tilde{p}_c = 0$ for all $c \in C$. As above, $V(\tilde{p}, \tilde{\mathcal{M}}) = V(p, \mathcal{M})$, and since $p$ is optimal on $A$, $\tilde{p}$ is optimal on $\tilde{A}$. In particular, $\tilde{p}_i = p_i > 0$, so $i$ still appears after cloning. $\qquad\square$

*Proof of Theorem 4.* We give a counterexample for $m = 3$. This suffices for all $m \geq 3$ since one may embed any $3 \times 3$ margin matrix into an $m \times m$ one by adding $m - 3$ robustly dominated alternatives. Let $A = \{1, 2, 3\}$. Remember that

$$v(p, M) = \min_{a \in A} p^\top M e_a.$$

Consider the three skew-symmetric matrices

$$M^a = \begin{pmatrix} 0 & -1 & -1 \\ 1 & 0 & -1 \\ 1 & 1 & 0 \end{pmatrix}, \quad M^b = \begin{pmatrix} 0 & -1 & -1 \\ 1 & 0 & 1 \\ 1 & -1 & 0 \end{pmatrix}, \quad M^c = \begin{pmatrix} 0 & 0 & 1 \\ 0 & 0 & 0 \\ -1 & 0 & 0 \end{pmatrix}.$$

Let $\mathcal{M}_1 := \mathrm{conv}\{M^a, M^b\}$ and $\mathcal{M}_2 := \mathrm{conv}\{M^a, M^c\}$.

Consider $p^\star := \left(0, \frac{1}{2}, \frac{1}{2}\right)$. We will show it is the robust maximal lottery for both sets $\mathcal{M}_1$ and $\mathcal{M}_2$.

Since $\mathcal{M}_1$ is the convex hull of two points and $v(p, \cdot)$ is concave, we have that

$$V(p, \mathcal{M}_1) = \min\{v(p, M^a), v(p, M^b)\}.$$

for every $p$. A direct computation gives

$$p^{\star\top} M^a = (1, \tfrac{1}{2}, -\tfrac{1}{2}), \qquad p^{\star\top} M^b = (1, -\tfrac{1}{2}, \tfrac{1}{2}),$$

so $v(p^\star, M^a) = v(p^\star, M^b) = -\frac{1}{2}$, hence

$$V(p^\star, \mathcal{M}_1) = -\frac{1}{2}. \tag{9}$$

Now let $p = (p_1, p_2, p_3) \in \Delta(A)$. Note that

$$p^\top M^a e_3 = -p_1 - p_2 = p_3 - 1, \qquad p^\top M^b e_2 = -p_1 - p_3 = p_2 - 1.$$

Therefore $v(p, M^a) \le p_3 - 1$ and $v(p, M^b) \le p_2 - 1$, hence

$$V(p, \mathcal{M}_1) = \min\{v(p, M^a), v(p, M^b)\} \le \min\{p_3 - 1, \; p_2 - 1\}.$$

If $p_3 \le \frac{1}{2}$ then $p_3 - 1 \le -\frac{1}{2}$. Otherwise $p_3 > \frac{1}{2}$ implies $p_2 < \frac{1}{2}$, hence $p_2 - 1 < -\frac{1}{2}$.

In either case, $V(p, \mathcal{M}_1) \le -\frac{1}{2}$ for all $p$. Together with Equation (9), this shows $p^\star \in \mathrm{RL}(\mathcal{M}_1)$. Following the same arguments, we have that $p^\star \in \mathrm{RL}(\mathcal{M}_2)$.

Let $\lambda = \frac{1}{2}$ and $\mathcal{M}_{1/2} := \frac{1}{2}\mathcal{M}_1 \oplus \frac{1}{2}\mathcal{M}_2$. Because $\mathcal{M}_1 = \mathrm{conv}\{M^a, M^b\}$ and $\mathcal{M}_2 = \mathrm{conv}\{M^a, M^c\}$, the set $\mathcal{M}_{1/2}$ is the convex hull of the four matrices

$$M^a, \quad \frac{M^a + M^b}{2}, \quad \frac{M^a + M^c}{2}, \quad \frac{M^b + M^c}{2}.$$

Fix $p$. Since $v(p, \cdot)$ is concave, $V(p, \mathcal{M}_{1/2})$ is the minimum of $v(p, M)$ over these four extreme points.

First, $M^a \in \mathcal{M}_{1/2}$, so

$$V(p^\star, \mathcal{M}_{1/2}) \le v(p^\star, M^a) = -\frac{1}{2}.$$

Next, define $p' := (0, \frac{1}{3}, \frac{2}{3})$. A direct computation gives:

$$v(p', M^a) = -\frac{1}{3}, \quad v\left(p', \frac{M^a + M^b}{2}\right) \ge -\frac{1}{3}, \quad v\left(p', \frac{M^a + M^c}{2}\right) \ge -\frac{1}{3}, \quad v\left(p', \frac{M^b + M^c}{2}\right) = -\frac{1}{3}.$$

Hence $V(p', \mathcal{M}_{1/2}) = -\frac{1}{3}$, and therefore

$$V(p', \mathcal{M}_{1/2}) = -\frac{1}{3} > -\frac{1}{2} \ge V(p^\star, \mathcal{M}_{1/2}).$$

Thus $p^\star \notin \mathrm{RL}(\mathcal{M}_{1/2})$.

We now prove the lower bound on the game value. Assume there exists $p^\star \in \mathrm{RL}(\mathcal{M}_1) \cap \mathrm{RL}(\mathcal{M}_2)$, so $V(p^\star, \mathcal{M}_k) = v^\star(\mathcal{M}_k)$ for $k = 1, 2$. Fix any $M_1 \in \mathcal{M}_1$ and $M_2 \in \mathcal{M}_2$. Then we have that

$$\min_{q \in \Delta(A)} p^{\star\top} (\lambda M_1 + (1 - \lambda) M_2) q \ge \lambda \min_{q \in \Delta(A)} p^{\star\top} M_1 q + (1 - \lambda) \min_{q \in \Delta(A)} p^{\star\top} M_2 q.$$

Taking $\min$ over $M_1 \in \mathcal{M}_1$ and $M_2 \in \mathcal{M}_2$ yields

$$V(p^\star, \mathcal{M}_\lambda) \geq \lambda V(p^\star, \mathcal{M}_1) + (1 - \lambda)V(p^\star, \mathcal{M}_2) = \lambda v^\star(\mathcal{M}_1) + (1 - \lambda)v^\star(\mathcal{M}_2).$$

Finally, since $v^\star(\mathcal{M}_\lambda) = \max_{p \in \Delta(A)} V(p, \mathcal{M}_\lambda)$, we have $v^\star(\mathcal{M}_\lambda) \geq V(p^\star, \mathcal{M}_\lambda)$, proving

$$v^\star(\mathcal{M}_\lambda) \geq \lambda v^\star(\mathcal{M}_1) + (1 - \lambda)v^\star(\mathcal{M}_2).$$

$\square$

*Proof of Theorem 5.* Fix $M \in \mathcal{M}_\lambda$. By Definition 9, there exist $M_1 \in \mathcal{M}_1$ and $M_2 \in \mathcal{M}_2$ such that $M = \lambda M_1 + (1-\lambda)M_2$. For every $j \in A$, since $i^\star$ is an RCW for both $\mathcal{M}_1$ and $\mathcal{M}_2$, we have $(M_1)_{i^\star j} \geq 0$ and $(M_2)_{i^\star j} \geq 0$. Hence

$$M_{i^\star j} = \lambda(M_1)_{i^\star j} + (1 - \lambda)(M_2)_{i^\star j} \geq 0.$$

Since this holds for all $j \in A$ and all $M \in \mathcal{M}_\lambda$, then $i^\star$ is an RCW for $\mathcal{M}_\lambda$. Applying Theorem 1 yields $e_{i^\star} \in \mathrm{RL}(\mathcal{M}_\lambda)$.

Under the additional assumption, Theorem 1 implies that $e_{i^\star}$ is the unique robust lottery for each $\mathcal{M}_k$, $k \in \{1, 2\}$. Thus $\mathrm{RL}(\mathcal{M}_1) = \mathrm{RL}(\mathcal{M}_2) = \{e_{i^\star}\}$, so any $p \in \mathrm{RL}(\mathcal{M}_1) \cap \mathrm{RL}(\mathcal{M}_2)$ must equal $e_{i^\star}$. Since $e_{i^\star} \in \mathrm{RL}(\mathcal{M}_\lambda)$, this proves the stated implication. $\square$

*Proof of Theorem 6.* Fix $p \in \Delta(A)$ and write $M(w) := \sum_{k=1}^K w_k M^{(k)}$. The robust value of $p$ under a weight set $\mathcal{W}$ is

$$V(p, \mathcal{W}) = \min_{w \in \mathcal{W}} \min_{a \in A} p^\top M(w) e_a.$$

We introduce an epigraph variable $t$ and rewrite the robust maximization as

$$\max_{p \in \Delta(A)} V(p, \mathcal{W}(w_0, \rho))$$
$$= \max_{p \in \Delta(A),\, t \in \mathbb{R}} \left\{ t : t \leq \min_{w \in \mathcal{W}(w_0, \rho)} p^\top M(w) e_a \; \forall a \in A \right\}.$$

Fix an alternative $a \in A$ and a lottery $p \in \Delta(A)$. Define the coefficients

$$c_{a,k}(p) := p^\top M^{(k)} e_a \qquad (k = 1, \ldots, K),$$

so that $p^\top M(w) e_a = \sum_{k=1}^K w_k c_{a,k}(p)$. Consider the inner minimization problem

$$\Phi_a(p) := \min_{w \in \Delta(K)} \sum_{k=1}^K w_k c_{a,k}(p) \quad \text{s.t.} \quad \frac{1}{2} \|w - w_0\|_1 \leq \rho.$$

Write the TV constraint with slack variables $s \in \mathbb{R}_{\geq 0}^K$:

$$w_k - w_{0,k} \leq s_k, \quad -(w_k - w_{0,k}) \leq s_k, \quad \sum_{k=1}^K s_k \leq 2\rho.$$

Thus $\Phi_a(p)$ is the optimal value of the linear program

$$\min_{w,s} \quad \sum_{k=1}^K w_k\, c_{a,k}(p)$$
$$\text{s.t.} \quad \sum_{k=1}^K w_k = 1, \quad w_k \geq 0 \;\; \forall k,$$
$$w_k - w_{0,k} \leq s_k, \quad -(w_k - w_{0,k}) \leq s_k \;\; \forall k,$$
$$\sum_{k=1}^K s_k \leq 2\rho, \quad s_k \geq 0 \;\; \forall k.$$

This LP is feasible (e.g. $w = w_0$, $s = 0$) and bounded (since $w$ lies in a compact polytope), so strong duality holds. A standard duality computation yields the dual

$$\max_{\mu_a, \lambda_a, \gamma_a} \quad \mu_a - 2\rho\,\lambda_a + \sum_{k=1}^{K} w_{0,k}\gamma_{a,k}$$
$$\text{s.t.} \quad \mu_a + \gamma_{a,k} \le c_{a,k}(p) \quad \forall k,$$
$$-\lambda_a \le \gamma_{a,k} \le \lambda_a \quad \forall k,$$
$$\lambda_a \ge 0,$$

where $\mu_a \in \mathbb{R}$ is free and $\gamma_a = (\gamma_{a,1}, \ldots, \gamma_{a,K})$. By strong duality, $\Phi_a(p)$ equals the optimum of this dual. Therefore, the robust problem $\max_{p \in \Delta(A)} \min_{a \in A} \Phi_a(p)$ is equivalently written as the single LP

$$\max_{p,t,\mu,\lambda,\gamma} \quad t$$
$$\text{s.t.} \quad \sum_{i \in A} p_i = 1, \quad p_i \ge 0 \;\; \forall i,$$
$$t \le \mu_a - 2\rho\,\lambda_a + \sum_{k=1}^{K} w_{0,k}\gamma_{a,k} \quad \forall a \in A,$$
$$\mu_a + \gamma_{a,k} \le p^\top M^{(k)} e_a \quad \forall a \in A,\ \forall k \in [K],$$
$$-\lambda_a \le \gamma_{a,k} \le \lambda_a \quad \forall a \in A,\ \forall k \in [K],$$
$$\lambda_a \ge 0 \quad \forall a \in A.$$

All constraints are linear in the variables, and the number of variables is $O(mK)$. $\qquad\square$

*Proof of Theorem 7.* Assume $\rho$ is *small* in the sense that

$$\rho \;\le\; \rho_0 \;:=\; \min\Big\{\min_{k \in [K]} w_{0,k},\; 1 - \max_{k \in [K]} w_{0,k}\Big\}. \tag{10}$$

Let $\pi \in \arg\max_{p \in \Delta(A)} V(p, \mathcal{W}(w_0, \rho))$, so $V(\pi, \mathcal{W}(w_0, \rho)) = v^\star$. Sample $I_1, \ldots, I_s \sim \pi$ i.i.d. and define

$$p := \frac{1}{s}\sum_{t=1}^{s} e_{I_t} \in \Delta(A), \qquad \text{so} \qquad |\operatorname{supp}(p)| \le s.$$

For each alternative $j \in A$ and group $k \in [K]$, define

$$f_{j,k}(r) := \sum_{i \in A} r_i\,M_{ij}^{(k)} \qquad (r \in \Delta(A)).$$

Define the $w_0$-average matrix $\bar{M} := \sum_{k=1}^{K} w_{0,k} M^{(k)}$ and

$$\bar{f}_j(r) := \sum_{i \in A} r_i\,\bar{M}_{ij} = \sum_{k=1}^{K} w_{0,k} f_{j,k}(r), \quad R_j(r) := \max_{k \in [K]} f_{j,k}(r) - \min_{k \in [K]} f_{j,k}(r).$$

Fix $r \in \Delta(A)$ and $j \in A$ and set $s_k := f_{j,k}(r) \in [-1, 1]$. Then

$$\min_{w \in \mathcal{W}(w_0, \rho)} \sum_{k=1}^{K} w_k f_{j,k}(r) \;=\; \bar{f}_j(r) \;-\; \rho\,R_j(r). \tag{11}$$

*Proof.* Write $w = w_0 + u$ with $\sum_k u_k = 0$ and $\|u\|_1 \le 2\rho$. For any constant $c$, $u^\top s = u^\top(s - c\mathbf{1})$, hence by Hölder,

$$u^\top s \ge -\|u\|_1\,\|s - c\mathbf{1}\|_\infty.$$

Choosing $c = \frac{1}{2}(\max_k s_k + \min_k s_k)$ gives $\|s - c\mathbf{1}\|_\infty = \frac{1}{2}(\max s - \min s) = \frac{1}{2}R_j(r)$, so $u^\top s \geq -(2\rho)(R_j(r)/2) = -\rho R_j(r)$. Thus $\min_w w^\top s \geq w_0^\top s - \rho R_j(r) = \bar{f}_j(r) - \rho R_j(r)$.

For achievability, let $k^+ \in \arg\max_k s_k$ and $k^- \in \arg\min_k s_k$ and define $\tilde{w} := w_0 - \rho e_{k^+} + \rho e_{k^-}$. Under Equation (10), we have $w_{0,k^+} \geq \rho$ and $w_{0,k^-} \leq 1 - \rho$, hence $\tilde{w} \in \Delta(K)$ and $\frac{1}{2}\|\tilde{w} - w_0\|_1 = \rho$.

Therefore $\tilde{w}^\top s = w_0^\top s - \rho(\max s - \min s) = \bar{f}_j(r) - \rho R_j(r)$, proving Equation (11). $\diamond$

Since $p^\top M(w)q$ is linear in $q$, we have $\min_{q \in \Delta(A)} r^\top M(w)q = \min_{j \in A} \sum_k w_k f_{j,k}(r)$. Moreover, because the expression is linear in $w$, the minimizations commute:

$$V(r, \mathcal{W}(w_0, \rho)) = \min_{j \in A} \min_{w \in \mathcal{W}(w_0,\rho)} \sum_k w_k f_{j,k}(r).$$

Applying Equation (11) yields the decomposition

$$V(r, \mathcal{W}(w_0, \rho)) = \min_{j \in A} \left[ \bar{f}_j(r) - \rho R_j(r) \right]. \tag{12}$$

Since $\bar{M}_{ij} \in [-1, 1]$, Hoeffding implies for each $j \in A$,

$$\Pr\left( \bar{f}_j(p) \leq \bar{f}_j(\pi) - \frac{\varepsilon}{2} \right) \leq \exp\left( -\frac{s\varepsilon^2}{8} \right).$$

A union bound over $j \in A$ gives

$$\Pr\left( \exists j : \ \bar{f}_j(p) \leq \bar{f}_j(\pi) - \frac{\varepsilon}{2} \right) \leq m \exp\left( -\frac{s\varepsilon^2}{8} \right). \tag{13}$$

If $\rho = 0$, then $R_j(\cdot)$ does not appear in Equation (12) and we may skip this step. Assume $\rho > 0$ and set $\eta := \varepsilon/(4\rho)$. For each $(j, k) \in A \times [K]$, Hoeffding gives the two-sided bound

$$\Pr\left( \left| f_{j,k}(p) - f_{j,k}(\pi) \right| \geq \eta \right) \leq 2 \exp\left( -\frac{s\eta^2}{2} \right).$$

A union bound over $(j, k)$ yields

$$\Pr\left( \exists (j, k) : \ \left| f_{j,k}(p) - f_{j,k}(\pi) \right| \geq \eta \right) \leq 2mK \exp\left( -\frac{s\eta^2}{2} \right). \tag{14}$$

Choose $s$ such that the right-hand sides of Equation (13) and Equation (14) are at most $1/4$. Equivalently, it suffices that

$$s \geq \frac{8}{\varepsilon^2} \log(4m) \qquad \text{and} \qquad s \geq \frac{2}{\eta^2} \log(8mK) = \frac{32\rho^2}{\varepsilon^2} \log(8mK).$$

Then with probability at least $1/2$ we have simultaneously for all $j \in A$,

$$\bar{f}_j(p) \geq \bar{f}_j(\pi) - \frac{\varepsilon}{2}, \tag{15}$$

and for all $(j, k) \in A \times [K]$,

$$\left| f_{j,k}(p) - f_{j,k}(\pi) \right| \leq \eta. \tag{16}$$

From Equation (16), for each $j$,

$$\max_k f_{j,k}(p) \leq \max_k f_{j,k}(\pi) + \eta, \qquad \min_k f_{j,k}(p) \geq \min_k f_{j,k}(\pi) - \eta,$$

hence $R_j(p) \leq R_j(\pi) + 2\eta$. Combining with Equation (15) gives, for each $j$,

$$\bar{f}_j(p) - \rho R_j(p) \ \geq \ \bar{f}_j(\pi) - \rho R_j(\pi) - \frac{\varepsilon}{2} - 2\rho\eta \ = \ \bar{f}_j(\pi) - \rho R_j(\pi) - \varepsilon,$$

since $2\rho\eta = \varepsilon/2$. Taking $\min_{j \in A}$ and using Equation (12) yields

$$V(p, \mathcal{W}(w_0, \rho)) \ \geq \ V(\pi, \mathcal{W}(w_0, \rho)) - \varepsilon \ = \ v^\star - \varepsilon.$$

Because the success event has positive probability, there exists a realization of the sample for which $|\operatorname{supp}(p)| \leq s$ and $V(p, \mathcal{W}(w_0, \rho)) \geq v^\star - \varepsilon$. $\qquad\square$

**Lemma A.1** (Concentration of empirical mixture in $\ell_1$)**.** *Let $w^\star \in \Delta(K)$ denote the (unknown) population mixture. We observe i.i.d. group labels $Z_1, \ldots, Z_n \sim w^\star$ and form $\hat{w} \in \Delta(K)$. Fix $\varepsilon \in (0,1)$ and $\delta \in (0,1)$. If*

$$n \;\geq\; \max\left\{ \frac{K}{\varepsilon^2}, \frac{2}{\varepsilon^2} \log\left( \frac{2}{\delta} \right) \right\},$$

*then with probability at least $1 - \delta$,*

$$\|\hat{w} - w^\star\|_1 \;\leq\; 2\varepsilon,$$

*and equivalently $d(\hat{w}, w^\star) := \frac{1}{2}\|\hat{w} - w^\star\|_1 \leq \varepsilon$.*

*Proof.* Let $N_k := \sum_{t=1}^n \mathbf{1}\{Z_t = k\}$. Then $\hat{w}_k = N_k/n$ and $N_k$ follows a binomial distribution with parameters $(n, w^\star_k)$. Denote $\Delta := \hat{w} - w^\star$.

Using $\mathbb{E}|\Delta_k| \leq \sqrt{\mathbb{E}[\Delta_k^2]} = \sqrt{\mathrm{Var}(\hat{w}_k)}$ and summing over $k$, we have that

$$\mathbb{E}\|\Delta\|_1 \;\leq\; \sum_{k=1}^K \sqrt{\mathrm{Var}(\hat{w}_k)} \;=\; \sum_{k=1}^K \sqrt{\frac{w^\star_k(1 - w^\star_k)}{n}} \;\leq\; \sum_{k=1}^K \sqrt{\frac{w^\star_k}{n}} \;=\; \frac{1}{\sqrt{n}} \sum_{k=1}^K \sqrt{w^\star_k} \;\leq\; \sqrt{\frac{K}{n}},$$

where the last step uses Cauchy–Schwarz inequality. Then, if $n \geq \frac{K}{\varepsilon^2}$, then $\mathbb{E}[\|\hat{w} - w^\star\|_1] \leq \varepsilon$.

Now define $f(Z_1, \ldots, Z_n) := \|\hat{w} - w^\star\|_1$. Replacing a single sample $Z_t$ by $Z'_t$ changes the value by at most $\frac{2}{n}$:

$$\left| f(Z_1, \ldots, Z_t, \ldots, Z_n) - f(Z_1, \ldots, Z'_t, \ldots, Z_n) \right| \leq \frac{2}{n} \quad \forall\, t \in [n].$$

By McDiarmid's inequality, for any $t > 0$,

$$\mathbb{P}(|f - \mathbb{E}f| \geq t) \;\leq\; 2\exp\left( -\frac{2t^2}{\sum_{i=1}^n (2/n)^2} \right) \;=\; 2\exp\left( -\frac{nt^2}{2} \right).$$

Then, if $n \geq \frac{2}{\varepsilon^2} \log \frac{2}{\delta}$, then $\|\hat{w} - w^\star\|_1 \leq 2\varepsilon$ with probability at least $1 - \delta$. We get our lemma by putting together both results. $\qquad\square$

**Lemma A.2** (Lipschitz continuity in the mixture weights)**.** *For any fixed $p \in \Delta(A)$ and any $w, w' \in \Delta(K)$,*

$$|v(p, w) - v(p, w')| \;\leq\; \|w - w'\|_1.$$

*Proof.* Fix $p \in \Delta(A)$ and $w, w' \in \Delta(K)$. Denote $\phi_w(q) := p^\top M(w)q$. For any $q \in \Delta(A)$, we have that:

$$|\phi_w(q) - \phi_{w'}(q)| = \left| p^\top \big( M(w) - M(w') \big)q \right| = \left| \sum_{k=1}^K (w_k - w'_k)\, p^\top M^{(k)} q \right| \leq \sum_{k=1}^K |w_k - w'_k|\, \|M^{(k)}\|_\infty \;\leq\; \|w - w'\|_1,$$

where we used $|p^\top X q| \leq \|X\|_\infty$ for $p, q \in \Delta(A)$. Taking the minimum over $q$ and using $|\min_q a_q - \min_q b_q| \leq \sup_q |a_q - b_q|$, we conclude that:

$$|v(p, w) - v(p, w')| = \left| \min_q \phi_w(q) - \min_q \phi_{w'}(q) \right| \;\leq\; \sup_{q \in \Delta(A)} |\phi_w(q) - \phi_{w'}(q)| \;\leq\; \|w - w'\|_1.$$

$\qquad\square$

**Lemma A.3** (Oracle inequality)**.** *Fix $\rho \geq 0$ and suppose $w^\star \in \mathcal{W}(\hat{w}, \rho)$. Then we have that $v(\hat{p}, w^\star) \geq \hat{v}$ and $v(\hat{p}, w^\star) \geq -4\rho$.*

*Proof.* Since $w^\star \in \mathcal{W}(\hat{w}, \rho)$, we have

$$v(\hat{p}, w^\star) \geq \min_{w \in \mathcal{W}(\hat{w}, \rho)} v(\hat{p}, w) = \hat{v}.$$

By optimality of $\hat{p}$,

$$\hat{v} = \min_{w \in \mathcal{W}(\hat{w}, \rho)} v(\hat{p}, w) \geq \min_{w \in \mathcal{W}(\hat{w}, \rho)} v(p^\star, w),$$

where $p^\star \in \arg\max_p v(p, w^\star)$ achieves $v(p^\star, w^\star) = v^\star = 0$. For any $w \in \mathcal{W}(\hat{w}, \rho)$, Lemma A.2 and the triangle inequality give

$$v(p^\star, w) \geq v(p^\star, w^\star) - \|w - w^\star\|_1 \geq 0 - \left( \|w - \hat{w}\|_1 + \|\hat{w} - w^\star\|_1 \right) \geq -\left(2\rho + 2\rho\right) = -4\rho,$$

since $w \in \mathcal{W}(\hat{w}, \rho)$ and $w^\star \in \mathcal{W}(\hat{w}, \rho)$ imply

$$\|w - \hat{w}\|_1 \leq 2\rho, \qquad \|\hat{w} - w^\star\|_1 \leq 2\rho.$$

Therefore,

$$\min_{w \in \mathcal{W}(\hat{w}, \rho)} v(p^\star, w) \geq -4\rho \implies \hat{v} \geq -4\rho.$$

This gives us that $v(\hat{p}, w^\star) \geq \hat{v} \geq -4\rho$. □

*Proof of Theorem 8.* By Lemma A.1, with probability at least $1 - \delta$,

$$\|\hat{w} - w^\star\|_1 \leq 2\rho,$$

which implies $d(\hat{w}, w^\star) \leq \rho$, i.e., $w^\star \in \mathcal{W}(\hat{w}, \rho)$. On this event, Lemma A.3 gives

$$\text{Regret}(w^\star) = -v(\hat{p}, w^\star) \leq 4\rho.$$

The final bound follows by substituting the definition of $\rho$. □

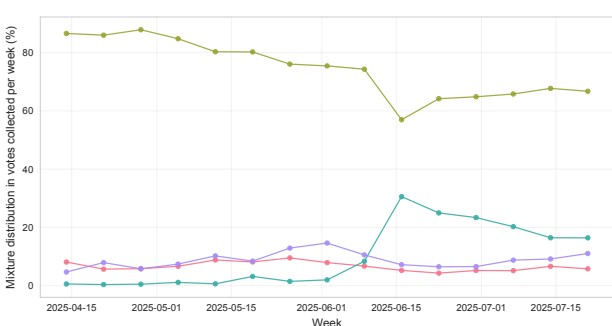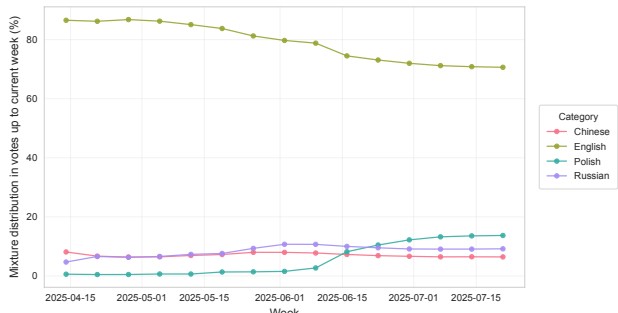

*Figure B.1.* Weekly (left) and cumulative (right) mixture distributions of vote categories in LMArena. The weekly plot shows the fraction of votes collected per category each week, while the cumulative plot shows the category composition of all votes collected up to a given week.

# B. Additional Experiments

All the following experiments were conducted on CPU using open-source datasets. We provide the code here. All results are reported with standard errors estimated from 200 bootstrap resamples, using an 80/20 train–test split of the benchmark.

## B.1. LMArena

LMArena (Chiang et al., 2024) is a large-scale, crowdsourced benchmark for evaluating language models via pairwise comparisons. Users interact with a randomly selected pair of models on prompts they submit, view the two responses side-by-side, and cast a vote for the better response. Because prompts are user-generated and interactions occur in the wild, the resulting preferences reflect a diverse mix of tasks, languages, and user populations, making the benchmark well-suited for studying heterogeneity and robustness in preference-based evaluation.

In particular, we use the most recent public release from LMArena[3]. The dataset contains approximately 136K pairwise votes collected on the LMArena platform between April 17, 2025 and July 25, 2025, spanning 53 frontier models. For our analysis, we restrict to a subset of 23 models to ensure sufficient head-to-head votes across model pairs. Moreover, we add a Laplace smoothing constant of $\eta = 1$ to regularize against small counts. Table B.1 specifies the models in consideration, along with their price in USD per 1M input tokens. These prices were sourced from the model provider if available or from AWS Bedrock otherwise.

Since annotators submit their votes anonymously and no demographic metadata is provided, prompt language is the only available group identifier. We define subpopulations by the language of the user-authored prompt and focus on the four most frequent prompt languages in this release—English, Polish, Russian, and Chinese. We present the mixture distribution of these languages in the LMArena data in Figure B.1 and specify the count of votes per category under our subset of models in Table B.8. We illustrate in Figure B.2 the preference heterogeneity in the LMArena dataset using the reversal method described in Section 5.1, along with specific examples in Table B.2.

---

[3]https://huggingface.co/datasets/lmarena-ai/arena-human-preference-140k

*Table B.1.* Models in LMArena and their input-token prices (USD per 1M input tokens). The symbol (T) designates a thinking version.

| Model | Price in USD per 1M input tokens |
| --- | ---: |
| GPT-4o v3 | 2.50 |
| GPT-4.1 Mini | 0.40 |
| o3 | 2.00 |
| o4 Mini | 1.10 |
| Haiku 3.5 | 0.80 |
| Sonnet 3.5 v2 | 3.00 |
| Sonnet 3.7 | 3.00 |
| Sonnet 3.7 (T) | 3.00 |
| Sonnet 4 | 3.00 |
| Sonnet 4 (T) | 3.00 |
| Opus 4 | 15.00 |
| Opus 4 (T) | 15.00 |
| DeepSeek R1 | 0.28 |
| DeepSeek V3 | 0.28 |
| Gemini 2.0 Flash | 0.10 |
| Gemini 2.5 Flash | 0.30 |
| Gemini 2.5 Pro | 1.25 |
| Gemma 3 | 0.23 |
| Llama 4 Maverick Exp | 0.24 |
| Llama 4 Maverick | 0.24 |
| Mistral Medium | 0.40 |
| Qwen3 235B | 0.22 |
| Qwen3 30B | 0.15 |

*Table B.2.* **Highest reversal-rate pairs in LMArena.** For each model pair $(i, j)$, we report the reversal probability and the languages which prefer $i$ over $j$, along with the number of comparisons available in each language (ZH = Chinese, PL = Polish, RU = Russian, EN = English).

| Pair $(i, j)$ | Reversal Probability | Prefer $i$ | Prefer $j$ |
| --- | --- | --- | --- |
| Gemini 2.0 Flash, Opus 4 | 0.467 | ZH (5), PL (17), RU (10) | EN (54) |
| Llama 4 Maverick, Gemma 3 | 0.453 | ZH (8), PL (13), RU (13) | EN (64) |
| Mistral Medium, Llama 4 Maverick | 0.443 | EN (127) | ZH (19), PL (30), RU (14) |
| o4 Mini, Gemini 2.5 Pro | 0.419 | EN (115) | ZH (10), PL (28), RU (11) |
| Qwen3 30B, Sonnet 3.7 | 0.415 | ZH (10), PL (20) | EN (64), RU (8) |

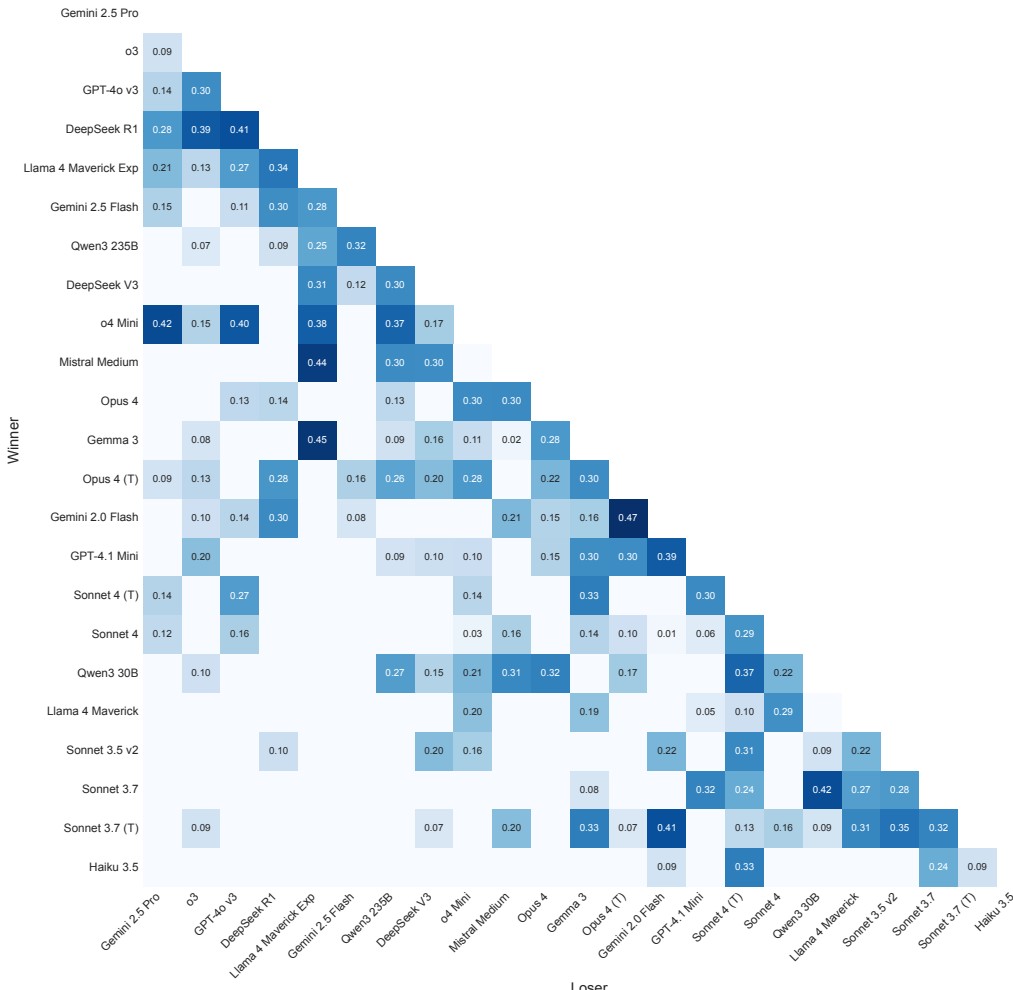

*Figure B.2.* **Pairwise preferences exhibit substantial heterogeneity across groups in LMArena.** We compute the probability that two distinct groups (sampled proportional to their comparison counts) yield opposite preferences for $(i, j)$, i.e., their margins $M_{ij}$ have opposite signs.

### B.2. HUMAINE Dataset

The HUMAINE leaderboard (Prolific AI, 2025) contains 174K pairwise preference votes comparing 33 large language models (LLMs). Each vote records the two models shown, the winner, and annotator metadata, including age, country of residence, ethnicity, and political affiliation. We focus on ethnicity as an axis of preference heterogeneity, and we also present results for political affiliation. To ensure that comparisons are sufficiently connected within each group, we restrict to the following set of 15 models, using $\eta = 1$ Laplace smoothing: Claude 3.7 Sonnet, Claude Opus 4, Claude Sonnet 4, Command A, Command R7B, Gemini 2.0 Flash, Gemini 2.5 Flash, Gemini 2.5 Pro, o1, o3 Mini, o4 Mini, Grok 3, Grok 4, DeepSeek R1, and Mistral Nemo.

### B.3. Search Arena

Search Arena (Miroyan et al., 2025) is a large-scale, crowdsourced benchmark for evaluating language models via pairwise comparisons on the task of web search. We use the most recent public release from Search Arena[4]. The dataset contains approximately 24K pairwise votes collected on the LMArena platform between March 18, 2025 and May 8, 2025, spanning 13 frontier models. We consider the full set of models, with a Laplace smoothing constant of $\eta = 1$ to regularize against small counts: Gemini 2.0 Flash, Gemini 2.5 Flash, Gemini 2.5 Pro, Gemini 2.5 Pro (no search), GPT-4o Mini, GPT-4o,

---

[4]https://huggingface.co/datasets/lmarena-ai/search-arena-24k

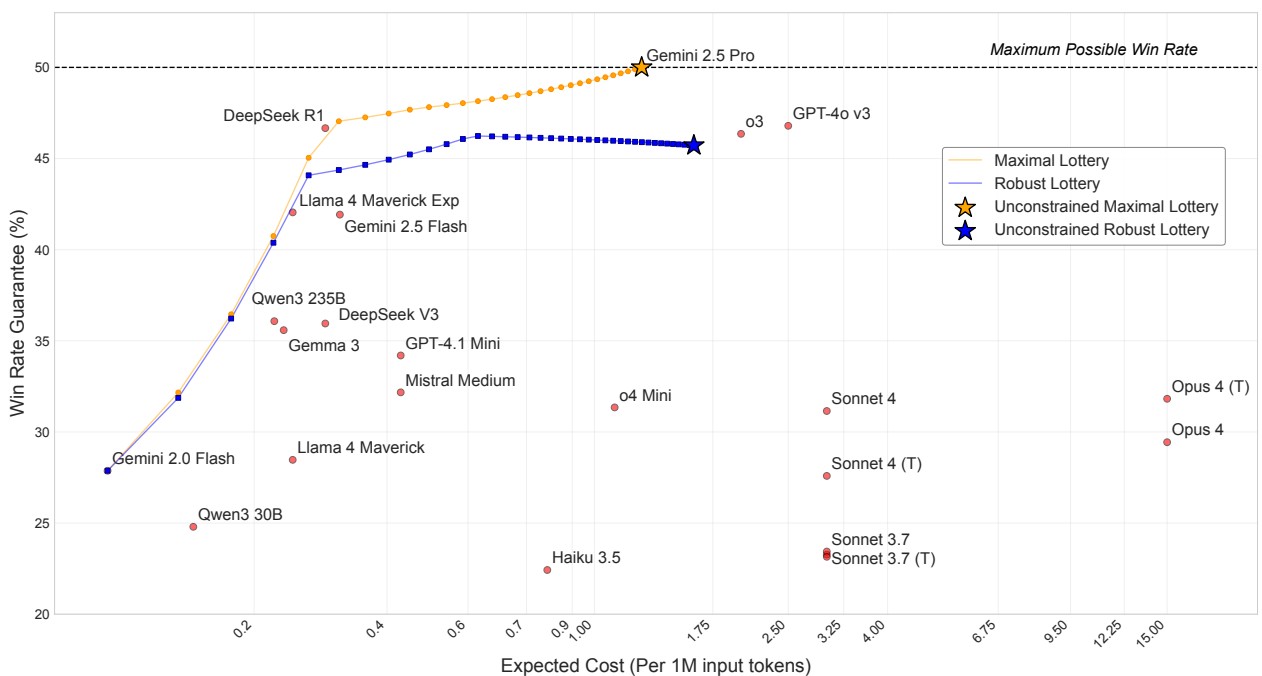

*Figure B.3.* Win rate-cost frontier on the overall preference matrix.

*Table B.3.* Held-out win rates on HUMAINE by ethnic group.

| Model / Mixture | Asian | Black | Mixed | White | Overall |
|---|---|---|---|---|---|
| Gemini 2.5 Pro | $0.38 \pm 0.01$ | $0.23 \pm 0.00$ | $0.10 \pm 0.00$ | $\mathbf{0.50} \pm 0.00$ | $0.49 \pm 0.00$ |
| Grok 3 | $0.40 \pm 0.00$ | $\mathbf{0.43} \pm 0.00$ | $0.17 \pm 0.00$ | $0.41 \pm 0.00$ | $0.44 \pm 0.00$ |
| Grok 4 | $0.23 \pm 0.01$ | $0.20 \pm 0.00$ | $0.19 \pm 0.01$ | $0.42 \pm 0.00$ | $0.43 \pm 0.00$ |
| Maximal Lottery ($\rho = 0$) | $0.40 \pm 0.00$ | $0.30 \pm 0.01$ | $0.21 \pm 0.01$ | $0.48 \pm 0.00$ | $\mathbf{0.50} \pm 0.00$ |
| Robust Lottery ($\rho = 0.8$) | $\mathbf{0.43} \pm 0.00$ | $0.42 \pm 0.00$ | $\mathbf{0.41} \pm 0.00$ | $0.42 \pm 0.00$ | $0.45 \pm 0.00$ |

GPT-4o High, GPT-4o High Loc, Sonar, Sonar Pro, Sonar Pro High, Sonar Reasoning, and Sonar Reasoning Pro High. We present results based on grouping the votes by prompt language or by the annotator's primary intent for the given task in the prompt. We filter out rows with non-unique or null group values.

## B.4. Open LLM

The Open LLM Leaderboard (Aidar Myrzakhan, 2024) includes responses from diverse model families on a set of standard LLM benchmarks, such as MMLU (Hendrycks et al., 2021) and ARC (Chollet et al., 2026). We evaluate the following model families: Claude, Gemini, Mistral, Cerebras, Gemma, GPT Neo, GPT 3.5, Llama 3, Olmo, OPT, Phi 3, Pythia, and Qwen. We compute preference matrices per benchmark as follows: for a given prompt, model $i$ wins over model $j$ if model $i$'s response is correct while model $j$'s response is incorrect.

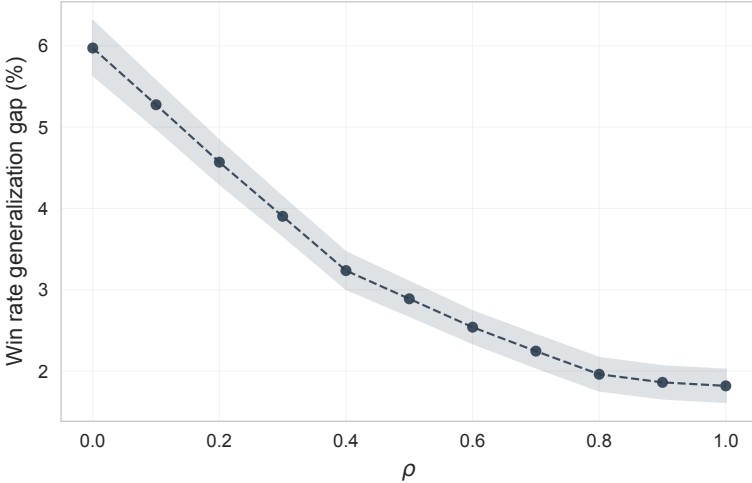

*Figure B.4.* **Robust lotteries reduce overfitting in LMArena.** We compute robust lotteries for varying radius values $\rho$ and evaluate each lottery on the overall population's training votes and held-out votes from LMArena. We report the generalization gap (train win rate minus test win rate) as a function of $\rho$. As $\rho$ increases, the gap shrinks from roughly $6\%$ at $\rho = 0$ – which corresponds to a standard maximal lottery – to about $2\%$ at $\rho = 1$, indicating that robustness regularizes the learned mixture and yields more stable performance.

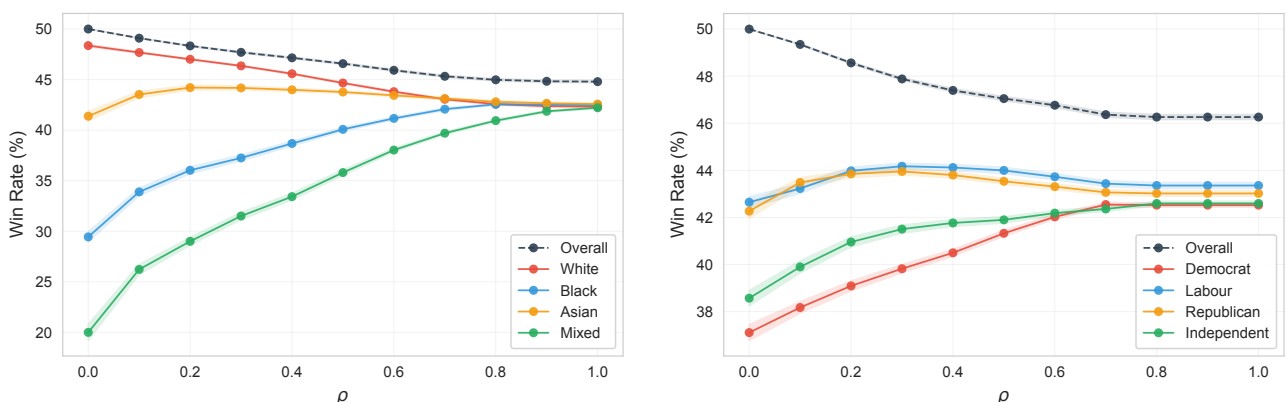

*Figure B.5.* **Win rate achieved by robust lottery on training margin matrices in the HUMAINE dataset.** Groups defined by annotator's ethnic group (**Left**) and political affiliation (**Right**).

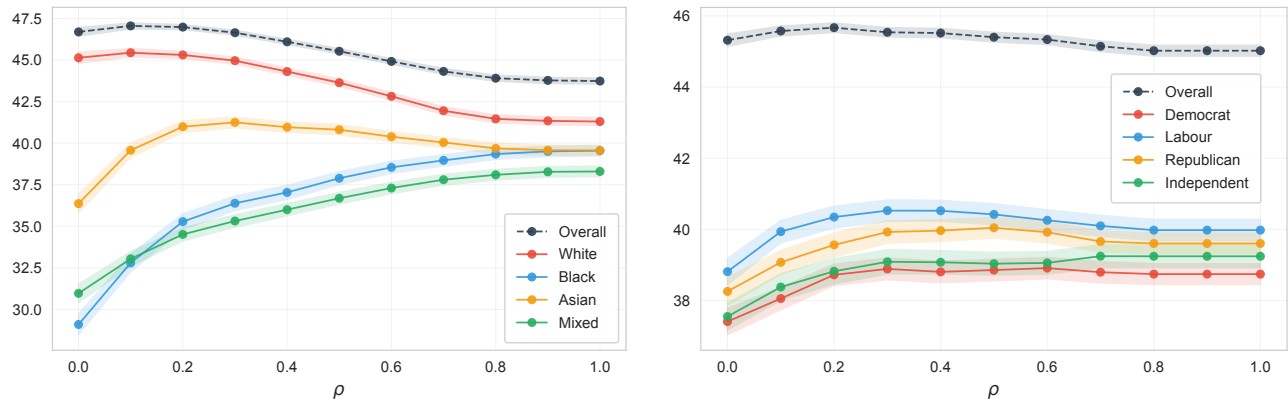

*Figure B.6.* **Win rate achieved by robust lottery on held-out votes in the HUMAINE dataset.** Groups defined by annotator's ethnic group (**Left**) and political affiliation (**Right**).

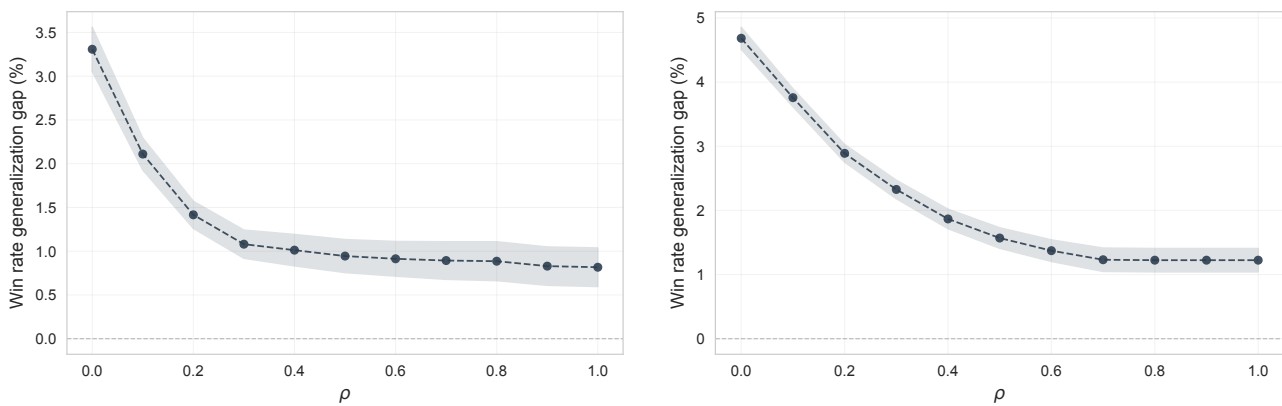

*Figure B.7.* **Robust lotteries reduce overfitting in the HUMAINE dataset.** Groups defined by annotator's ethnic group (**Left**) and political affiliation (**Right**).

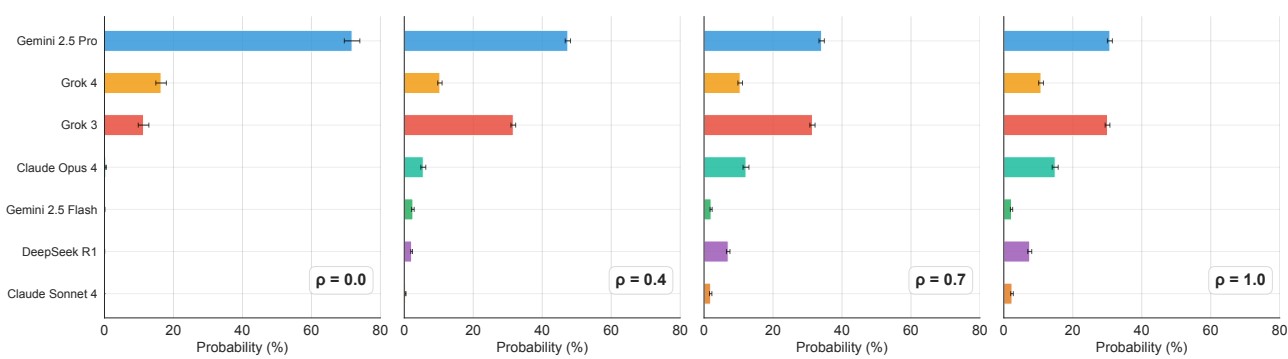

*Figure B.8.* **Robust lotteries diversify the lottery to handle preference tradeoffs among ethnic groups in the HUMAINE dataset.** We present the estimated probability assigned to each model when annotators are grouped based on ethnic group.

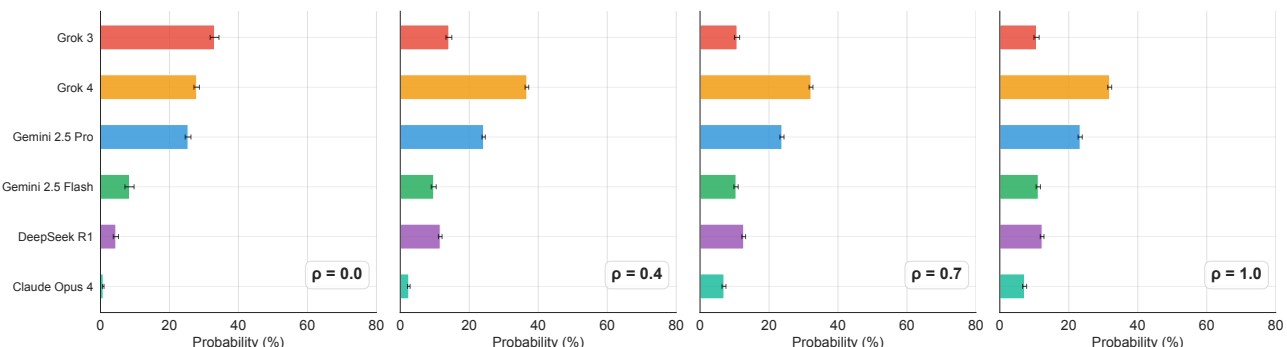

*Figure B.9.* **Robust lotteries diversify the lottery to handle preference tradeoffs among political groups in the HUMAINE dataset.** Groups defined by political affiliation.

*Table B.4.* Held-out win rates on HUMAINE by political group.

| Model / Mixture | Democrat | Independent | Labour | Republican | Overall |
|---|---|---|---|---|---|
| Gemini 2.5 Flash | $0.19 \pm 0.01$ | $0.30 \pm 0.01$ | $0.40 \pm 0.01$ | $0.27 \pm 0.01$ | $0.39 \pm 0.00$ |
| Gemini 2.5 Pro | $\mathbf{0.44} \pm 0.00$ | $0.14 \pm 0.01$ | $0.27 \pm 0.01$ | $0.38 \pm 0.01$ | $0.36 \pm 0.00$ |
| Grok 3 | $0.22 \pm 0.01$ | $0.29 \pm 0.01$ | $0.24 \pm 0.01$ | $0.31 \pm 0.01$ | $0.40 \pm 0.00$ |
| Grok 4 | $0.13 \pm 0.01$ | $0.31 \pm 0.01$ | $0.31 \pm 0.01$ | $0.29 \pm 0.01$ | $0.38 \pm 0.00$ |
| Maximal Lottery ($\rho = 0$) | $0.36 \pm 0.00$ | $0.38 \pm 0.00$ | $0.42 \pm 0.00$ | $0.43 \pm 0.00$ | $\mathbf{0.50} \pm 0.00$ |
| Robust Lottery ($\rho = 0.8$) | $0.42 \pm 0.00$ | $\mathbf{0.42} \pm 0.00$ | $\mathbf{0.43} \pm 0.00$ | $\mathbf{0.43} \pm 0.00$ | $0.46 \pm 0.00$ |

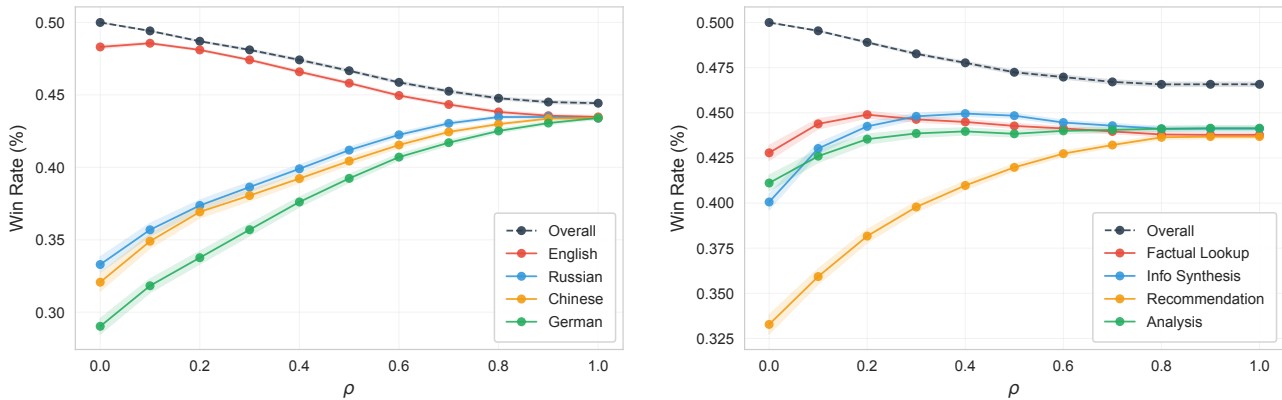

*Figure B.10.* **Win rate achieved by robust lottery on training margin matrices in the Search Arena dataset.** Groups defined by annotator's prompt language (**Left**) and primary intent (**Right**).

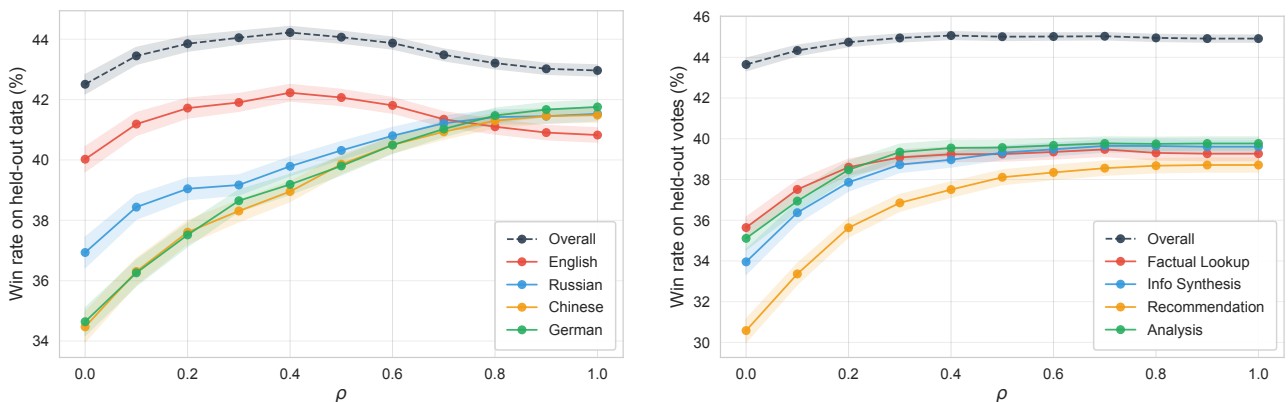

*Figure B.11.* **Win rate achieved by robust lottery on held-out votes in the Search Arena dataset.** Groups defined by annotator's prompt language (**Left**) and primary intent (**Right**).

*Table B.5.* Held-out win rates on Search Arena by language group.

| Model / Mixture | Chinese | English | German | Russian | Overall |
|---|---|---|---|---|---|
| Gemini 2.5 Pro Grounding | $0.29 \pm 0.01$ | $0.46 \pm 0.01$ | $0.23 \pm 0.01$ | $0.30 \pm 0.01$ | $0.47 \pm 0.00$ |
| Sonar | $0.16 \pm 0.00$ | $0.39 \pm 0.00$ | $0.18 \pm 0.01$ | $0.30 \pm 0.01$ | $0.41 \pm 0.00$ |
| Sonar Pro | $0.15 \pm 0.00$ | $0.34 \pm 0.01$ | $0.27 \pm 0.01$ | $0.24 \pm 0.01$ | $0.36 \pm 0.00$ |
| Maximal Lottery ($\rho = 0$) | $0.32 \pm 0.01$ | $\mathbf{0.48} \pm 0.00$ | $0.28 \pm 0.01$ | $0.33 \pm 0.01$ | $\mathbf{0.50} \pm 0.00$ |
| Robust Lottery ($\rho = 1.0$) | $\mathbf{0.43} \pm 0.00$ | $0.44 \pm 0.00$ | $\mathbf{0.43} \pm 0.00$ | $\mathbf{0.43} \pm 0.00$ | $0.44 \pm 0.00$ |

*Table B.6.* Held-out win rates on Search Arena by primary intent.

| Model / Mixture | Analysis | Factual Lookup | Info Synthesis | Recommendation | Overall |
|---|---|---|---|---|---|
| Gemini 2.5 Pro Grounding | $0.36 \pm 0.01$ | $0.41 \pm 0.01$ | $0.38 \pm 0.01$ | $0.28 \pm 0.01$ | $0.46 \pm 0.00$ |
| Sonar Pro High | $0.30 \pm 0.01$ | $0.24 \pm 0.01$ | $0.26 \pm 0.01$ | $0.43 \pm 0.01$ | $0.43 \pm 0.00$ |
| Sonar Reasoning Pro High | $0.39 \pm 0.01$ | $0.41 \pm 0.01$ | $0.35 \pm 0.01$ | $0.29 \pm 0.01$ | $0.47 \pm 0.00$ |
| Maximal Lottery ($\rho = 0$) | $0.41 \pm 0.01$ | $0.42 \pm 0.01$ | $0.39 \pm 0.01$ | $0.34 \pm 0.01$ | $\mathbf{0.50} \pm 0.00$ |
| Robust Lottery ($\rho = 0.9$) | $\mathbf{0.44} \pm 0.00$ | $\mathbf{0.44} \pm 0.00$ | $\mathbf{0.44} \pm 0.00$ | $\mathbf{0.44} \pm 0.00$ | $0.47 \pm 0.00$ |

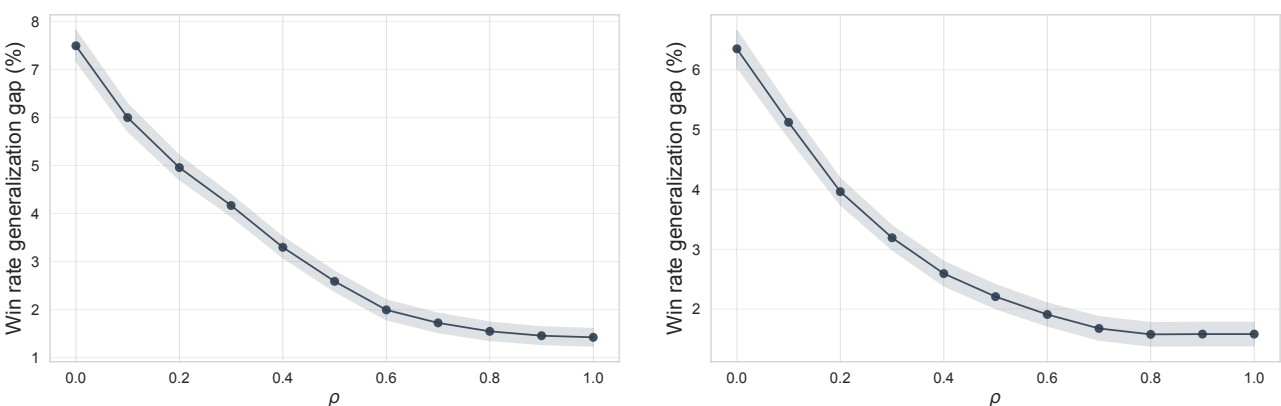

*Figure B.12.* **Robust lotteries reduce overfitting in the Search Arena dataset.** Groups defined by annotator's prompt language (**Left**) and primary intent (**Right**).

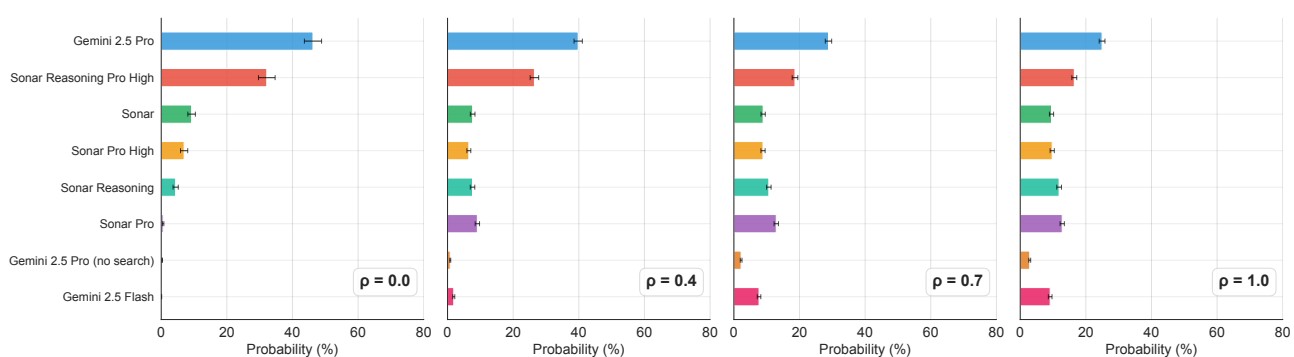

*Figure B.13.* **Robust lotteries diversify the lottery to handle preference tradeoffs among languages in the Search Arena dataset.** Groups defined by prompt language.

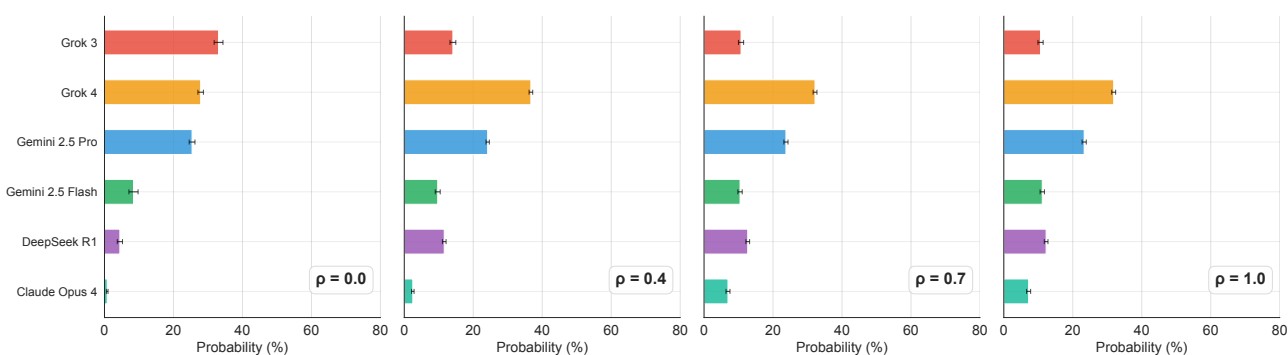

*Figure B.14.* **Robust lotteries diversify the lottery to handle preference tradeoffs among tasks in the Search Arena dataset.** Groups defined by the annotator's primary intent in the task.

*Table B.7.* Held-out win rates by benchmark in Open LLM.

| Model / Mixture | ARC | CommonsenseQA | MedMCQA | MMLU | Overall |
|---|---|---|---|---|---|
| Claude | $0.31 \pm 0.00$ | $0.32 \pm 0.01$ | $\mathbf{0.50} \pm 0.00$ | $\mathbf{0.50} \pm 0.00$ | $\mathbf{0.49} \pm 0.00$ |
| Mistral | $\mathbf{0.48} \pm 0.00$ | $0.41 \pm 0.01$ | $0.36 \pm 0.00$ | $0.46 \pm 0.00$ | $0.49 \pm 0.00$ |
| GPT 3.5 | $0.47 \pm 0.00$ | $\mathbf{0.47} \pm 0.00$ | $0.33 \pm 0.00$ | $0.37 \pm 0.00$ | $0.45 \pm 0.00$ |
| Maximal Lottery ($\rho = 0$) | $0.33 \pm 0.01$ | $0.33 \pm 0.01$ | $0.47 \pm 0.00$ | $0.49 \pm 0.00$ | $0.49 \pm 0.00$ |
| Robust Lottery ($\rho = 0.7$) | $0.40 \pm 0.00$ | $0.38 \pm 0.00$ | $0.42 \pm 0.00$ | $0.45 \pm 0.00$ | $0.48 \pm 0.00$ |

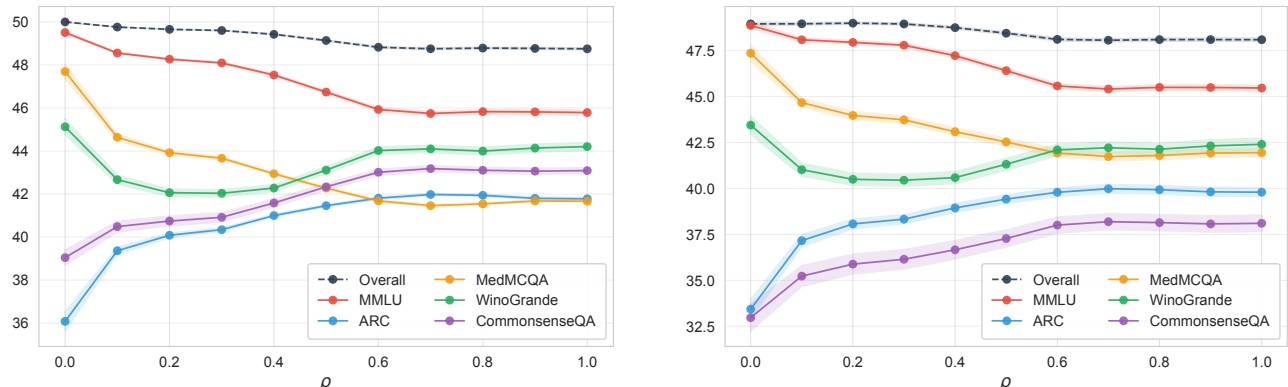

*Figure B.15.* Groups defined by **benchmark** on Open LLM. **Left:** Win rate achieved on training margin matrices. **Right:** Win rate achieved on held-out margin matrices.

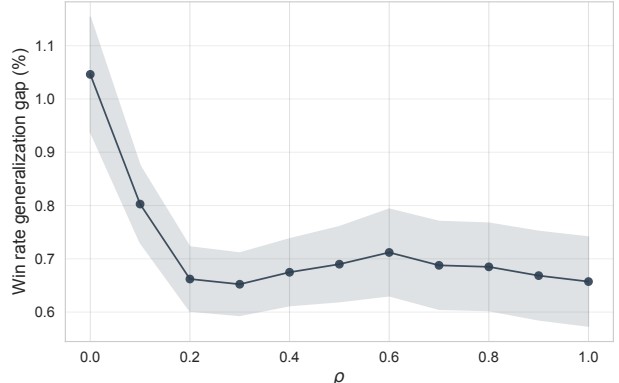

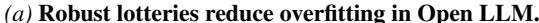

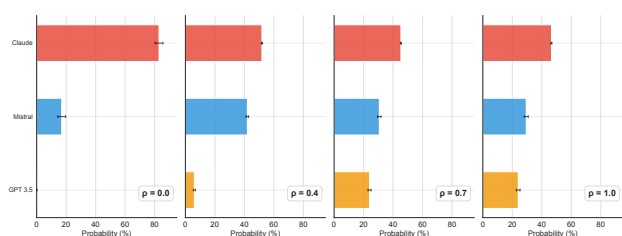

*(a)* **Robust lotteries reduce overfitting in Open LLM.**

*(b)* **Robust lotteries diversify the lottery to handle preference tradeoffs among languages in the Open LLM dataset.**

*Table B.8.* Vote counts across all benchmarks.

| Benchmark | Dimension | Group | Votes |
|---|---|---|---|
| LMArena | Language | English | 26,172 |
| | | Polish | 4,842 |
| | | Russian | 3,530 |
| | | Chinese | 2,565 |
| HUMAINE | Ethnicity | White | 14,266 |
| | | Black | 4,134 |
| | | Asian | 3,359 |
| | | Mixed | 1,392 |
| | Political | Democrat | 3,740 |
| | | Labour | 3,594 |
| | | Republican | 2,998 |
| | | Independent | 2,826 |
| Search Arena | Language | English | 12,849 |
| | | Russian | 2,604 |
| | | Chinese | 1,150 |
| | | German | 637 |
| | Intent | Factual Lookup | 4,269 |
| | | Information Synthesis | 3,931 |
| | | Recommendation | 2,441 |
| | | Analysis | 2,348 |
| Open LLM | Benchmark | MMLU | 226,540 |
| | | ARC | 100,061 |
| | | MedMCQA | 47,767 |
| | | WinoGrande | 41,227 |
| | | CommonsenseQA | 19,303 |

