# OpenReview forum: "Robust AI Evaluation through Maximal Lotteries"
_ICML.cc/2026/Conference — ICML 2026 regular_

### Official Review · Reviewer_ZZ7b · 2026-03-05

**Soundness:** 4
**Presentation:** 2
**Significance:** 2
**Originality:** 3
**Overall Recommendation:** 4
**Confidence:** 4

**Summary:**

This paper proposes robust lotteries to evaluate LLM evaluation. As current evaluation strategies assume single ranking, it may be non transitive or cannot represent diverse quality (e.g. language). This paper solves that problem with robust lottery for evaluation.

**Compliance With Llm Reviewing Policy:**

Affirmed.

**Final Justification:**

Thank the authors for their efforts to address my concerns.

Through the rebuttal, the authors suggested that operational constraints could be incorporated into their framework; however, their "respectful disagreement" still fails to fully acknowledge the significant practical burden of maintaining multiple models in production. Furthermore, their argument that an evaluation methodology should remain independent of model optimization is a convenient academic defense that fails to address the demand for model improvement.

Nevertheless, I recognize the effort put into the rebuttal and the potential of the proposed non-transitive, robust evaluation paradigm as a timely theoretical contribution. Therefore, I have decided to increase my score to Weak Accept, as the work offers a meaningful, albeit idealized, framework for rethinking LLM leaderboards.

**Key Questions For Authors:**

- Sorry for my possible misunderstanding, but if the maximal lotteries assume pairwise margin and not assume transitivity, why should we deviate from pairwise comparison methods like bradley terry model?
- How can I trust this evaluation is reliable? How can I measure which evaluation is more trustful?
- Even with this robust lotteries, if some new data is given, its evaluation would be unreliable if the data have never seen before (not even given as worst case)?
- maximal lottery is the strategy that maximizes its worst-case margin against any opponent mixture. Suppose when a strategy is equally preferred over a opponent. As groud truth, the margin should be 0. However, the maximal lottery strategy will maximize its margin, causing wrong learning direction. How can it be solved?

**Limitations:**

See above.

**Strengths And Weaknesses:**

- Good formulation and theoretical base.
- This paper solves an important problem of preference non-transitivity and applying the concept of maximal lottery is interesting.
- For presentation, I cannot understand what the authors wanted to address with the Maximal lotteries paragraph in section 2.
- After all, this paper is related to LLM evaluation. It seems the actual meaning is limited as it cannot change any model parameter and learning process.
- The axis of Figure 1 is worst case win rate; where is normal win rate? win rate over what? Which point should I focus on?
- I still cannot understand whether Robust lottery is better than standard evaluation metrics for evaluating the general performances of LLMs.

---

> ### Author Rebuttal · Authors · 2026-03-31
>
> Thank you for your feedback! We address your questions below and would be happy to answer any additional questions during the discussion period.
>
> ----
>
> ###  Q1: Why should we deviate from pairwise comparison methods like BT?
>
> Both BT and lotteries are evaluation methods that use pairwise preference data collected from users. However, they differ in how this data is **aggregated**.
>
> BT enforces strong assumptions on the data: it assumes pairwise outcomes come from a single latent score per model, which imposes a global ranking. This global ranking can be misleading when preferences are heterogeneous, as is common in LLM evaluation. Our method uses the same pairwise preference data, but aggregates it without imposing any assumptions on the data. The example in our response to Q4 illustrates this idea.
>
> In terms of practical benefit, we refer you to our response to Reviewer zyHN. We illustrate how practitioners can use robust lotteries, and we provide empirical evidence that robust lotteries outperforms BT.
>
> ---
>
> ### Q2: How can I trust this evaluation is reliable? How can I measure which evaluation is more trustful?
>
> The key advantage of our framework is that we optimize a quantity with a direct operational meaning: the win rate against the worst-case opponent model. This gives a clear criterion for comparing two models, or two mixtures of models. Accordingly, we show that robust lotteries empirically outperforms BT (see the answer to Reviewer zyHN).
>
> ---
>
> ### Q3: Even with these robust lotteries, if some new data is given, its evaluation would be unreliable if the data have never seen before (not even given as worst case)?
>
> Our framework offers a principled way to obtain better out-of-sample guarantees under a stated family of distribution shifts. While this guarantee does not extend to arbitrary new data outside the specified uncertainty set, the advantage of the framework is that the uncertainty set is explicit and flexible, so it can be enlarged to cover the kinds of shifts one expects in deployment. Moreover, our empirical results on various LLM benchmarks show that increasing robustness improves the performance on unseen data (see Figure 3).
>
> ----
>
> ### Q4: Suppose when a strategy is equally preferred over a opponent. As ground truth, the margin should be 0. However, the maximal lottery strategy will maximize its margin, causing wrong learning direction. How can it be solved?
>
>
> The key point is that the pairwise margins are fixed by the observed comparison data. If two models are equally preferred, their margin is 0 and the maximal-lottery procedure does not change that quantity. Instead it uses this margin matrix to select a distribution over models.
>
> Consider three models $(A, B, C)$ such that the observed pairwise preference probabilities are $P(A \succ B)=0.5$, $P(A \succ C)=1$, and $P(B \succ C)=1$. We obtain the margin matrix:
>
> \\[
> M = P - P^T =
> \\begin{pmatrix}
> 0 & 0 & 1 \\\\
> 0 & 0 & 1 \\\\
> -1 & -1 & 0
> \\end{pmatrix}.
> \\]
>
> For example, $M_{12} = P(A \succ B) - P(B \succ A) = 0.5 - 0.5 = 0$. Hence, models $A$ and $B$ are equally preferred.
>
> The maximal lottery uses this given matrix $M$ to choose a distribution over models. By solving the maximin problem described in Definition 2, the maximal lottery is:
>
> $$
> p^\star = \left(\frac{1}{2}, \frac{1}{2}, 0\right).
> $$
>
> Hence, the optimal strategy is to choose model $A$ with probability $1/2$, $B$ with probability $1/2$, and never choose $C$.
>
> If the concern is instead that the estimated margin may be slightly nonzero because of finite-sample noise, then we agree that this is an important issue; this motivates our robust extension which provides a performance guarantee over a set of possible margin matrices.
>
> ----
>
> ### I cannot understand what the authors wanted to address with the maximal lotteries paragraph in Section 2.
>
> In the related work section, we situate our problem within the social choice literature, which has extensively studied methods for aggregating pairwise preferences. Maximal lotteries are the natural starting point for our work, and we  wanted to highlight that this notion has been studied beyond social choice, including in game theory and reinforcement learning.
>
> ---
>
> ### The axis of Figure 1 is worst case win rate; where is normal win rate?
>
> For each model or lottery, we first compute its win rate on each subgroup, and then take the minimum win rate across subgroups. While individual models or maximal lotteries perform poorly for some groups, we show in Figure 1 that robust lotteries provide stronger guarantees across the user population. Moreover, Figure 7 in Appendix A reports the corresponding frontier plots for each group. These plots show that robust lotteries incur only a small reduction in overall performance, while yielding much stronger worst-case guarantees and better performance on held-out votes (Figure 3a).

---

> > ### Author Rebuttal · Reviewer_ZZ7b · 2026-04-03
> >
> > Thank the authors for their thorougb responses. Below are my further questions.
> >
> > 1. The authors suggest using the derived robust lottery as a "routing policy" to achieve worst-case win rate guarantees. However, from a practitioner's perspective, maintaining a "frontier set" of multiple models—as opposed to a single top-ranked model from a BT leaderboard—introduces significant operational overhead, including increased VRAM usage and service inconsistency.
> >
> > 2. The lack of direct impact on the model training process also seems as a weakness in terms of overall significance.

---

> > > ### Author Response · Authors · 2026-04-04
> > >
> > > Thank you for your continued engagement. We respectfully disagree with the current comments, as we explain below.
> > >
> > > ---
> > >
> > > > The authors suggest using the derived robust lottery as a "routing policy" to achieve worst-case win rate guarantees. However, from a practitioner's perspective, maintaining a "frontier set" of multiple models—as opposed to a single top-ranked model from a BT leaderboard—introduces significant operational overhead, including increased VRAM usage and service inconsistency.
> > >
> > > Model routing is an important aspect of AI deployment, particularly when models are only accessed via API. For instance, OpenRouter and Cursor allow easy model selection, and CLI coding tools (such as Gemini CLI and Codex) implement auto-routing to reduce operational cost. The practical advantage of our framework is that the operational constraints you highlighted, such as VRAM usage or service overhead, can be incorporated directly into the evaluation. This allows any practitioner to find the best set for their needs given such constraints.
> > >
> > > That said, it is important to emphasize that our main focus is evaluation, whereas routing is a potential downstream application. We will include these comments in our revised manuscript.
> > >
> > > ---
> > >
> > > > The lack of direct impact on the model training process also seems as a weakness in terms of overall significance.
> > >
> > > While our work focuses on evaluation and does not impact model training, this is not a weakness. In fact, an evaluation methodology *should not* be used directly to optimize models, as its validity as a measurement tool can degrade [1].
> > >
> > > Moreover, evaluation itself is a major area of interest at ICML [2] and across machine learning. The purpose of evaluation work is to provide an external criterion for comparing models and assessing their capabilities. Our contribution directly tackles limitations in existing evaluation frameworks amid recent calls to rethink LLM evaluation [3].
> > >
> > > ---
> > >
> > > [1] Dominguez-Olmedo, R., Dorner, F. E., & Hardt, M. Training on the Test Task Confounds Evaluation and Emergence. In The Thirteenth International Conference on Learning Representations.
> > >
> > > [2] https://icml.cc/Conferences/2026/CallForPapers
> > >
> > > [3] Singh, S., Nan, Y., Wang, A., D'Souza, D., Kapoor, S., Üstün, A., ... & Hooker, S. The Leaderboard Illusion. In The Thirty-ninth Annual Conference on Neural Information Processing Systems Datasets and Benchmarks Track.

---

### Official Review · Reviewer_nC8v · 2026-03-09

**Soundness:** 3
**Presentation:** 3
**Significance:** 3
**Originality:** 3
**Overall Recommendation:** 3
**Confidence:** 4

**Summary:**

The authors consider the problem of evaluating large language models using pairwise comparisons where there the answers from two LLMs are represented to a human or a strong LLM judge who would decide which answer is better or a tie when both answers are close. Using these pairwise comparisons, the existing methods based of win-rate and ELO rating ranks the models. In this paper, the authors instead output a subset of front-tier models using the pairwise preferences. The authors motivate that, using different user population for different subset of questions would cause distributional in variance and the existing methods do not account for such distribution shift.

The authors further point out that the existing methods do not account for clones (presence of more than one copy of the same family of models) that would affect the overall rankings of other models, and asymmetric data when the pairwise preferences result in cycle dependencies. In a way, robust lotteries account for all these shortcomings of the existing methods, instead of considering a single majority matrix of pairwise preferences it accounts for a set of plausible majority matrices and gives the optimal strategy accounting for the worst possible realization of the data.

I have checked the proof sketches in main paper, but only skimmed the proofs and experimental results in Appendix.

**Compliance With Llm Reviewing Policy:**

Affirmed.

**Final Justification:**

After the discussion and follow-ups with the authors, I feel that this work needs stronger empirical validation before it is ready for acceptance. I might have been more inclined to recommend acceptance if this were positioned primarily as a social choice theory paper, but as an LLM evaluation paper, the lack of detailed experiments and a clear comparison to exiting methods makes it difficult for me to recommend acceptance.

My reasons are two fold: (i) despite my attempts to follow up on specific concerns, I did not receive responses that sufficiently clarify the concerns; and (ii) I still find it difficult to interpret the proposed maximal-lottery outputs in a way that supports the kind of model comparison practitioners expect (e.g., understanding which model is “better” and by how much), and I am not yet convinced if this framework is appropriate for LLM evaluation without stronger experimental evidence.

Because the authors did not address my follow-up questions in detail, I spent additional time reading background material on maximal lotteries and how one could (if possible) interpret bipartisan set in practice. My understanding is that models in the bipartisan (frontier) set are not necessarily always better than all models outside it (this is because of cycles), which makes it unclear to me how a tiered representation and any implied ordering within and across tiers, should be interpreted. This is precisely where experiments would be most useful: implementing the proposed tiering (if intended), comparing it to existing ranking pipelines, and demonstrating that the outputs are both interpretable and practically meaningful.

Overall, I find the theoretical results on robust lotteries strong and very interesting. However, the application to LLM evaluation does not feel convincing in the absence of thorough experiments that demonstrate its practical use. Without such experiments, I worry the paper risks remaining primarily theoretical with limited actionable value for LLM evaluation in practice. I encourage the authors to incorporate these experiments in future revisions.

**Key Questions For Authors:**

Definition 1: In pairwise comparisons a human evaluator (or a strong LLM) can choose which models answer is better or indicate a tie. However, in the current formulation does not account for ties. This is particularly important because of how the authors motivate the necessity of a new methodology for evaluation to account for clones. When clones are present, it will most likely be a tie. Does incorporating ties to this methodology would change the analysis? can you elaborate on this?

Some prior work goes beyond producing a single rank and instead returns a set of plausible rankings to reflect uncertainty from limited or noisy comparisons, especially when some model pairs are rarely compared. Also there are methods that combine human and AI judges comparisons can output rank sets (a set of possible rankings a model can take) with statistical guarantees, that is, ensuring the returned rank set contains the true ranking when evaluation relies only on human pairwise comparisons. These sets will be larger when the uncertainity is high and a singleton when the method is more certain about a model's rank. It would be good if the authors noted that not all approaches output a single total order, some methods explicitly return subsets of possible rankings to account for data ambiguity (see, e.g., [1]).

[1] Prediction powered ranking of large language models.

In this regard, I have the following question. Can the earlier work on prediction-powered ranking be viewed as also addressing uncertainty in the comparison data (e.g., by combining a small set of human judgments with a larger set of LLM judgments)? If so, is it fair to characterize that framework as operating with an ambiguity set consisting of two (human and strong LLM) comparisons or margin matrices, whereas the current approach proposed by authors allows an ambiguity set containing an arbitrary number of matrices? If not, could you clarify the precise sense in which the two notions of uncertainty differ, and why the prediction-powered setting is not naturally captured in your formulation?

One thing that I am not sure about is how to interpret the output, as a practitioner may expect a (possible) ranking of models. For example, when the method returns a robust maximal lottery $p^*$, can this be viewed as a ranking, in the sense that $p_i$ reflects how much probability mass the robust equilibrium mixture assigns to model $i$? If so, is it reasonable to order models by $p_i$ and treat that ordering as a ranking?

The reason I ask is, the robust maximal lottery problem may admit multiple optimal solutions $p*$ (i.e., the solution need not be unique). In that case, ranking models by $p_i$ could yield different orderings across optimal lotteries. Would it be appropriate to summarize this by reporting, for each model, a range of possible positions (rank intervals), or more generally a set of plausible ranks induced by the set of optimal $p*$, as a way to communicate the ambiguity under distributional uncertainty? Is this interpretation consistent with the intended use of this method? Can you please clarify on how this methods can be incorporated to a benchmarking arena to rank the models?

I would like to further clarify on this aspect. I understand that the robust bipartisan set can be interpreted as a collection of frontier or top-performing models. However, is there a principled way to derive a ranking, or at least a rank set or rank intervals, from the robust-lottery output? Otherwise, it may be challenging to relate these results to common evaluation settings such as LLM ranking arenas. While I agree that under distributional uncertainty a single total order can be misleading, on the other hand reporting only a frontier (robust bipartisan) set may be hard to interpret for users accustomed to leaderboards. It could therefore be helpful to include an additional ranking style summary that reflects uncertainty, for example, tiered groupings or rank intervals induced by the set of optimal robust lotteries so readers can see not only which models lie on the frontier, but also how uncertain their relative ordering is. Can you clarify on this?

At the same time, I am somewhat skeptical that LLM ranking arenas will move away from presenting leaderboards or rankings in the near term. As a result, I worry that the proposed outputs, while theoretically well-motivated, may be harder to adopt in practice if they cannot be communicated in a ranking like format (this is ofcoarse in my point-of-view). It would therefore strengthen the paper if the authors either presented their results in a way that is more directly comparable to leaderboard style reporting, or discussed concrete ways to extend or post process the output into ranking style summaries (e.g., tiers, rank intervals).

**Limitations:**

See weaknesses and questions to authors.

**Strengths And Weaknesses:**

Condorcet consistency and clone in variance are well established properties of non-robust maximal lotteries. In this work, the authors extend these properties to robust maximal lotteries. In my opinion, these extensions are non-trivial, the paper thus dedicates a significant portion to formulate analogous properties for robust maximal lotteries and then give theoretical results establishing that robust lotteries satisfy them. This is the key technical contribution of this work. While the theoretical results are impressive, I found that some aspects of the experimental evaluation is not clear.

Since the proposed method outputs a set of frontier models and not the traditional way of ranking of models, it is not clear to me how to compare the output of this method with the existing methodologies. In particular, how inaccurate are existing methods based on rankings under these uncertainties and how much the proposed approach would change or improve the rankings in practice.

---

> ### Author Rebuttal · Authors · 2026-03-31
>
> Thank you for your thoughtful and detailed review! We address your comments below and would be happy to answer any additional questions during the discussion period.
>
> ---
>
> ### Q1: Does incorporating ties into this methodology change the analysis?
>
> Our framework can incorporate ties directly at the level of the estimated margin matrix. If, for a pair of models $(i,j)$, we observe $w_{ij}$ wins for $i$, $w_{ji}$ wins for $j$, and $t_{ij}$ ties, we can define the empirical margin as
> $$
> M_{ij} = \frac{w_{ij} - w_{ji}}{w_{ij} + w_{ji} + t_{ij}}.
> $$
> In other words, ties reduce the magnitude of the estimated margin, reflecting weaker evidence for a strict preference. The rest of the analysis is unchanged.
>
> ---
>
> ### Q2: Is it fair to characterize that the framework in [1] as operating with an ambiguity set consisting of two comparisons or margin matrices, whereas the current approach proposed by authors allows an ambiguity set containing an arbitrary number of matrices?
>
> You're correct that one could view the prediction-powered ranking framework as a way to generate an uncertainty set over pairwise preference matrices $M$ by combining the human and "strong LLM" comparisons. This can then be directly plugged into our proposed method to generate a robust lottery.
>
> However, our solution concept of producing a robust lottery has several advantages over the solution concept of producing subsets of possible rankings: First, our robust lottery produces a frontier set that has a theoretical guarantee of resistance to clones, which is not present in prior work on confidence sets over rankings. Second, our robust lottery can also be applied to produce a model selection policy with a guaranteed win rate over all other selection policies, which is outside the scope of prior work on producing confidence sets over rankings. We agree that the goal of handling uncertainty in evaluation is common to our work and this literature on robust rankings. We will add the above discussion to the related work section.
>
> ---
>
> ### Q3: Is there a principled way to derive a ranking, or at least a rank set or rank intervals, from the robust-lottery output?
>
> Thank you for the thoughtful question and suggestions! If one were to just apply the most straightforward extension of ranking models simply by their lottery selection probabilities, the resulting ranking would lose the clone-robustness property.
>
> In general, there is not an obvious way to convert a maximal lottery into a single ranking over models while preserving such properties.  Lanctot et al. (2023) (https://arxiv.org/pdf/2312.03121) have previously extended lotteries to produce a ranking over *sets* of models, which is still different from a conventional leaderboard. If leaderboard generation is the goal, our robust maximal lotteries can be plugged into the ranking generation algorithm from Lanctot et al. (2023) to produce a robust version of their rankings over sets. Therefore, we consider robust lotteries as a complementary view of AI evaluation. We refer you to our response to reviewer zyHN on the practical implications of robust lotteries.

---

> > ### Author Rebuttal · Reviewer_nC8v · 2026-04-01
> >
> > I thank the authors for their clarifications. I have a few follow-up questions, and some of my concerns still remain.
> >
> > 1. My concern regarding the practical applicability of the work still remains. As it stands, the contribution appears mostly theoretical, and its direct impact to evaluation of large language models (in ranking arenas) seems limited.
> >
> > 2. Can you please elaborate on the work of Lanctot et al. (2023) regarding generating single ranking via lotteries?
> >
> > (i) Is there a peer-reviewed version of this work (since the current citation is an arXiv preprint from 2023)?
> >
> > (ii) Where exactly in this paper is the transformation to rankings described? Given its length, a precise pointer (section) would be very helpful.
> >
> > (iii) A brief explanation of how these rankings are constructed, and how they differ from conventional ranking methods, would also be appreciated.
> >
> > (iv) Finally, could you clarify how your framework applies in this setting? Are any modifications required for your method to work with such ranking-based outputs?

---

> > > ### Author Response · Authors · 2026-04-02
> > >
> > > **Q1:** Thank you for the follow-up. We agree that rankings are the familiar interface in current LLM arenas. Our point is not to replace leaderboards with another ranking, but to provide a complementary benchmark that is robust to clone attacks and remains well defined under heterogeneous preferences. Importantly, **it can be computed from the same pairwise comparison data already collected by existing arenas**, so deploying our method is straightforward in practice. This is an important point and we will include a section describing the implementation of this in detail in the revised version. We also affirm that our theoretical contributions are a necessary first step to future practical implementations.
> > >
> > >
> > > ----
> > >
> > > **Q2:**
> > > (i) We do not believe that this work has been peer-reviewed.
> > >
> > > (ii) Please check Algorithm 1 (Iterative Maximal Lotteries) in Page 6. We apologize for not specifying a pointer earlier.
> > >
> > > (iii) The algorithm proceeds iteratively. Starting from the full set of alternatives, they compute a maximal lottery over the current set and identify the corresponding bipartisan set. The alternatives in this bipartisan set are then removed, yielding a reduced set. This procedure is repeated until no alternatives remain. The process induces a partition of the original set into successive tiers. The authors then opt to rank models within each tier according to their associated lottery weights. This differs from conventional ranking methods because it produces a ranking over *sets* of models, rather than a ranking over individual models.
> > >
> > > (iv) We can repeat the exact process to obtain tiers by computing robust lotteries instead of maximal lotteries.

---

### Official Review · Reviewer_yL7W · 2026-03-12

**Soundness:** 3
**Presentation:** 3
**Significance:** 2
**Originality:** 3
**Overall Recommendation:** 4
**Confidence:** 4

**Summary:**

Summary: The paper introduces robust lotteries as a way to evaluate models as an alternative to the BT framework and maximal lotteries, which can be sensitive to annotator or prompt heterogeneity. The paper provides an analysis proving properties of robust lotteries such as robust Condorcet consistency and weak-clone invariance and demonstrates improvements in worst-group test performance on real datasets.

**Compliance With Llm Reviewing Policy:**

Affirmed.

**Final Justification:**

The authors provided helpful clarifications. The paper provides a thorough analysis of robust lotteries and motivates them in terms of better accounting for diverse preferences. They demonstrate the utility of robust lotteries directly on leaderboards. While this is novel and advances evaluations, the method appears to have limited utility over BT models in downstream use cases. In particular, using a set of models at inference is less common and determining a frontier set of models can be done with standard rankings. However, it does address new aspects of evaluation, and therefore, I keep my score.

**Key Questions For Authors:**

- Could you explain how to interpret a set of models or utilize the resulting set in practice?

**Limitations:**

Yes

**Strengths And Weaknesses:**

Strengths:

- The paper introduces robust lotteries as a method for evaluating models which optimizes worst-case performance and under distribution shifts, provides more robust results than maximal lotteries, and better captures heterogeneous preferences, an important aspect of preference learning.
- The paper provides a thorough analysis of robust lotteries demonstrating that it has desirable properties as well as computational tractability under reasonable conditions.
- The paper demonstrates the benefits of robust lotteries on multiple leaderboards, and the results support the use of robust lotteries for better accounting for heterogeneity.
- The paper analyzes cost vs. performance, overfitting, and verifies heterogeneity of preferences across groups.

Weaknesses:

- While robust lotteries do improve worst-group test performance and identify a set of frontier models, it seems unclear how to interpret the set of models and use this set of models in practice for decision-making or deployment. Further discussion on this would be helpful.
- Figure 2 (b) seems a bit difficult to follow based on the example and (a) and (c). Is the mixed solution a mixture of the settings in (a) and (c) as this is what seems to be suggested in the main text?

---

> ### Author Rebuttal · Authors · 2026-03-31
>
> Thank you for your thoughtful feedback! We address below the questions and points raised in your review and would be happy to answer any additional questions.
>
> ---
>
> ### Q1/W1: Could you explain how to interpret a set of models or utilize the resulting set in practice?
>
> The support of the robust lottery identifies the frontier set, namely the models that remain competitive under the robust objective. The lottery weights indicate how strongly each model contributes to the worst-case guarantee.
>
> Our method has practical implications in two directions. We will add a subsection expanding on these directions in the revision.
>
> 1. **Evaluation:** Robust lotteries can be used to construct a benchmark of frontier models. As new models are released, the benchmark shows whether they enter the frontier and at what cost level. Unlike a single ranking, this can capture multiple models with complementary strengths.
>
> 2. **Model selection:** For practitioners, the frontier can be viewed as a principled shortlist. Rather than testing every available model, one can restrict attention to the frontier models at the relevant budget and then select among them using some application-specific criteria. Unlike BT, robust lotteries directly optimize for downstream performance: the win rate achieved against the strongest opponent model. If desired, this distribution can be used directly as a routing policy by sampling a model from the lottery per query.
>
> ----
>
> ### W2: Figure 2(b) seems a bit difficult to follow based on the example and (a) and (c). Is the mixed solution a mixture of the settings in (a) and (c) as this is what seems to be suggested in the main text?
>
> Your reading is  correct: panel (b) depicts the maximal lotteries induced by mixing the margin matrices from panels (a) and (c\). The main takeaway is that small changes in those mixture weights can move the maximal lottery noticeably. This motivates the need for a robust objective that provides strong guarantees under distribution shifts. We will clarity this point in the paper.

---

> > ### Author Rebuttal · Reviewer_yL7W · 2026-04-03
> >
> > Thank you for the response.
> >
> > Thank you for the confirmation on Figure 2, and the descriptions of how robust lotteries will be used for evaluation and model selection are helpful.

---

### Official Review · Reviewer_zyHN · 2026-03-13

**Soundness:** 3
**Presentation:** 3
**Significance:** 4
**Originality:** 4
**Overall Recommendation:** 5
**Confidence:** 4

**Summary:**

This paper presents an alternative evaluation protocol for LLM binary comparison data. The authors discuss how the existing BT models suffer from undesirable properties, potentially obfuscating interesting artifacts in the preference data, and imposing artificial structure. They then propose an augmented version of maximal lotteries, a well-established technique for social choice, proving useful properties and proposing a tractable algorithm to find the solution to the worst-case minimax game.

**Compliance With Llm Reviewing Policy:**

Affirmed.

**Final Justification:**

The authors addressed all my concerns, improving my assessment of the paper.

**Key Questions For Authors:**

- How is your model expected to be practically employed? As an AI engineer, should I use the derived distribution to sample models randomly during inference expecting to perform better than competitors even with the same prompt? As a researcher, should I use an ensemble with a mixture informed by the probabilities in the robust lottery? This is mentioned only in a single sentence in the concluding section.
- Why is your model practically better than BT? What are the main arguments you would use to convince someone maintaining leaderboards with BT to switch to your approach? Why are lotteries not immediately appropriate to derive leaderboards, and what would a system like this look like if not a leaderboard?
- You mention how BT derived latent scores, whereas your approach is "nonparametric"; to the best of my understanding, however, both techniques essentially result in a vector of equal length: the "ELO" and the probabilities of the lottery. Moreover, both methods depend on a few hyperparameters. Could you elaborate why these are different enough to warrant the distinction and the discussion in the main paper? I would appreciate a response focused on substance and not the semantics of a "parameter".

**Limitations:**

yes

**Strengths And Weaknesses:**

- Soundness: the reasoning for the choices made is technically sound, as well as intuitive. This was a very interesting and enlightening read. Theoretical results are elegant and seem robust, though I was unable to check everything and in detail. In terms of empirical results, while the authors set up their approach as a more robust, in various ways, evaluation technique compared to BT, they do not demonstrate in their results how it improves model comparison in practice. For example, in the abstract, the authors state that their approach has a better win-rate, but no BT empirical comparisons are made to the best of my understanding. Moreover, they do not articulate clearly and based on their empirical results what the main takeaway is for practitioners. Based on the results, which model should I deploy? The BT model directly offers a solution: the LLM with the highest latent score.
- Presentation: I was surprised by how easily I could follow the presentation of the theoretical results even from my first read, which speaks volumes to the effort the authors went to ensure good readability. As mentioned above, the final delivery and the main takeaways are lacking due to the one-dimensional empirical evaluation.
- Significance: the authors, while they make decent arguments in the introduction, do not fully flesh out how their approach could practically be applied to improve: model comparison, model selection, etc.
- Originality: this is an elegant and (at least intuitively) simple solution to a pertinent problem. The authors' extension of previous work is novel and catered to the problem at hand.

---

> ### Author Rebuttal · Authors · 2026-03-31
>
> Thank you for your helpful feedback and thoughtful reading of our work. We address below the questions and points raised in your review and would be happy to answer any additional questions.
>
> ---
>
> ### Q1: How is your model expected to be practically employed?
>
> Our method has practical implications in two directions. We agree that elaborating on these in the paper will be useful, and we will add a subsection expanding on these directions in the revision.
>
> 1. **Evaluation:** Robust lotteries can be used to construct a benchmark of frontier models. As new models are released, the benchmark shows whether they enter the frontier and at what cost level. Unlike a single ranking, this can capture multiple models with complementary strengths.
>
> 2. **Model selection:** For practitioners, the frontier can be viewed as a principled shortlist. Rather than testing every available model, one can restrict attention to the frontier models at the relevant budget and then select among them using some application-specific criteria. Unlike BT, robust lotteries directly optimize for downstream performance: the win rate achieved against the strongest opponent model. This distribution can be used directly as a routing policy by sampling a model from the lottery per query.
>
> ---
> ### Q2a:  Why is your model practically better than BT?
>
> Thank you for this question. We extend our results in Figure 3(a) on LMArena to show how robust lotteries provide stronger empirical guarantees than *any* algorithm that picks a single winner, including BT. We will include results on the remaining benchmarks in the revised version.
>
> We compare the robust lottery to the maximal lottery and to the strongest single-model baselines. The robust lottery achieved the best win rate across all subgroups on held-out votes. This is the practical distinction from BT-style leaderboards: BT is forced to select one winner, while robust lotteries optimize performance under heterogeneous preferences.
>
> | Model / Mixture | English | Polish | Chinese | Russian | Overall |
> |---|---:|---:|---:|---:|---:|
> | Deepseek R1 | 31.25 ± 0.70 | 24.76 ± 0.57 | 26.13 ± 0.52 | 26.17 ± 0.63 | 36.92 ± 0.50 |
> | Gemini-2.5 Pro | 40.97 ± 0.66 | 31.13 ± 0.85 | 29.48 ± 0.72 | 25.76 ± 0.54 | 44.22 ± 0.50 |
> | Llama 4 Maverick | 24.70 ± 0.70 | 25.53 ± 0.68 | 29.75 ± 0.77 | 25.80 ± 0.58 | 32.55 ± 0.56 |
> | Maximal Lottery | 40.78 ± 0.66 | 32.69 ± 0.83 | 32.06 ± 0.72 | 28.99 ± 0.68 | 44.06 ± 0.48 |
> | Robust Lottery ($\rho=0.4$) | **41.89** ± 0.44 | **38.01** ± 0.56 | **37.09** ± 0.54 | **35.71** ± 0.55 | **44.24** ± 0.32 |
>
> ----
>
> ### Q2b: What are the main arguments to convince someone maintaining leaderboards with BT to switch to your approach? Why are lotteries not immediately appropriate to derive leaderboards, and what would a system like this look like if not a leaderboard?
>
> This is a great point, and we will elaborate on it in the revised version. We view robust lotteries as both a complementary and a necessary view of AI evaluation. Robust lotteries provide stronger and more actionable guarantees than BT by answering the practical question: "what models should I deploy given heterogeneous preferences and diverse tasks?” We accomplish this while satisfying key properties that BT fails, such as being invariant to clones and not requiring restrictive preference assumptions. Since we do not return a ranking of models, we envision a system that reports the cost-performance frontier, showing which models are competitive at each budget as in Figure 1.
>
> If one were to just apply the most straightforward extension of ranking models simply by their lottery selection probabilities, the resulting ranking would lose the clone-robustness property. In general, there is not an obvious way to convert a maximal lottery into a single ranking over models while preserving such properties. Lanctot et al. (https://arxiv.org/pdf/2312.03121) have previously extended lotteries to produce a ranking over *sets* of models, which is still different from a conventional leaderboard. If leaderboard generation is the goal, our robust maximal lotteries can be plugged into the ranking generation algorithm from Lanctot et al. to produce a robust version of their rankings over sets.
>
> ----
>
> ### Q3: You mention how BT derived latent scores, whereas your approach is "nonparametric"... Could you elaborate why these are different enough to warrant the distinction and the discussion in the main paper?
>
> The distinction is not the dimensionality of the output, but the presence of a structural assumption on the preference model. BT assumes that each model has a latent score and pairwise outcomes are generated through a specific parametric form. Robust lotteries make no such assumption: they are computed directly from the observed majority-margin matrix. Similarly, the robustness parameter controls the decision criterion, but it does not impose a structural assumption on the preference model.

---

> > ### Author Rebuttal · Reviewer_zyHN · 2026-03-31
> >
> > The authors have adequately addressed all my concerns.

---

### Decision · Program_Chairs · 2026-04-30

**Decision:**

Accept (regular)

**Comment:**

This paper introduces robust lotteries for aggregating pairwise preferences under heterogeneity and distribution shift. The remaining concerns are mostly about practicality, interpretability, and lack of empirical comparison, but the core contributions and theory appears solid and novel, and reviewers appreciate that. I thus recommend acceptance.